# MVCustom: Multi-View Customized Diffusion via Geometric Latent Rendering and Completion

**Minjung Shin[1]**  **Hyunin Cho[1]**  **Sooyeon Go[1]**  **Jin-Hwa Kim[2,3]**  **Youngjung Uh[1]***

[1]Yonsei University  [2]NAVER AI Lab  [3]SNU AIIS

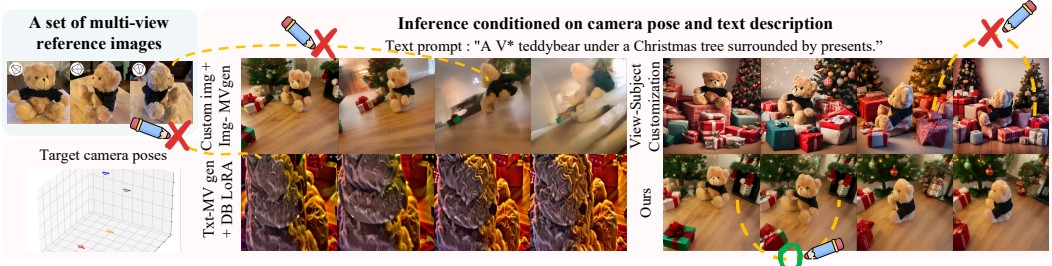

Figure 1: **Comparison between MVCustom and existing approaches extended to multi-view customization.** The light blue box shows the reference multi-view images and corresponding camera poses of a customized object. The 'X' marks indicate regions inconsistent with either the reference object's appearance or across views, while 'O' marks indicate well-maintained consistency. Our approach clearly outperforms existing methods by achieving accurate viewpoint alignment and robust multi-view consistency for both the customized object and novel surroundings generated from diverse textual prompts.

## Abstract

Multi-view generation with camera pose control and prompt-based customization are both essential elements for achieving controllable generative models. However, existing multi-view generation models do not support customization with geometric consistency, whereas customization models lack explicit viewpoint control, making them challenging to unify. Motivated by these gaps, we introduce a novel task, *multi-view customization*, which aims to jointly achieve multi-view camera pose control and customization. Due to the scarcity of training data in customization, existing multi-view generation models, which inherently rely on large-scale datasets, struggle to generalize to diverse prompts. To address this, we propose *MVCustom*, a novel diffusion-based framework explicitly designed to achieve both multi-view consistency and customization fidelity. In the training stage, MVCustom learns the subject's identity and geometry using a feature-field representation, incorporating the text-to-video diffusion backbone enhanced with dense spatio-temporal attention, which leverages temporal coherence for multi-view consistency. In the inference stage, we introduce two novel techniques: *depth-aware feature rendering* explicitly enforces geometric consistency, and *consistent-aware latent completion* ensures accurate perspective alignment of the customized subject and surrounding backgrounds. Extensive experiments demonstrate that *MVCustom* achieves the most balanced and consistent competitive performance across multi-view consistency, customization fidelity, demonstrating effective solution of multi-objective generation task. Project page: https://minjung-s.github.io/mvcustom/

| Task | Method | Fidelity | Holistic | S.MV | H.MV |
|------|--------|----------|----------|------|------|
| (a) Customization | DreamBooth, CustomDiffusion, etc. | O | O | X | X |
| (b) Subject-only text-to-MV gen. | FlexGen, Make-Your-3D, etc. | X | X | O | X |
| (c) Text-to-MV generation | CameraCtrl, ViewDiff, etc. | X | O | O | O |
| (d) Subject-only image-to-MV gen. | SV3D, SyncDreamer, etc. | X | X | O | X |
| (e) Image-to-MV gen. | SEVA, CAT3D, ViewCrafter, etc. | X | O | O | O |
| (f) Viewpoint-aware subject custom. | CustomDiffusion360, CustomNet | O | O | O | X |
| **(g) Multi-view customization** | **MVCustom (ours)** | O | O | O | O |

Table 1: **Comparison of existing tasks and representative methods.** *Fidelity* refers to preserving object identity from reference images and alignment with textual prompts in customization. *Holistic* denotes whether both subjects and the surroundings described in a prompt are synthesized. *S.MV* evaluates whether subjects remain consistent across different viewpoints. *H.MV* consistency refers to whether both subjects and their surroundings are holistically consistent across viewpoints. *MV* stands for multi-view.

# 1    INTRODUCTION

As generative models advance rapidly, users are increasingly demanding fine-grained controllability. Among the essential elements, two forms of control are significant: camera control and customization. First, *camera control* is to generate images for specified viewpoints, which is essential in domains such as 3D understanding. In particular, ensuring camera pose control and multi-view consistency for both the subject and its surroundings is crucial for realistic and immersive content, as misalignment across views severely undermines geometric coherence. Second, *customization* is to capture user-specific subjects, or concepts, supporting personalized content generation and supporting applications such as creative media and design prototyping, *etc*.

While each form of control is valuable on its own, integrating them unlocks significantly richer applications. A unified framework that supports both capabilities enables 3D customization for virtual prototyping and personalized asset generation, where both user-specific fidelity and geometric consistency are indispensable. Moreover, it broadens the scope of controllable generative models, enabling realistic, immersive, and user-tailored content beyond the reach of existing approaches. To this end, we introduce the novel task of *multi-view customization*, which requires (1) generating images that adhere to specified camera parameters for consistent perspective alignment, (2) preserving subject identity provided by reference images, and (3) coherently adapting both subjects and their surrounding context to diverse textual prompts.

However, to the best of our knowledge, no prior method fully satisfies the requirements of the multi-view customization. As summarized in Tbl. 1, conventional customization methods (Lee et al., 2024; Ruiz et al., 2023; Kumari et al., 2024) preserve reference identity and align with prompts, but lack viewpoint control. Most multi-view generation methods focus only on subjects, neglecting consistent surroundings across views (cases b, d in Tbl. 1). Some holistic multi-view generation methods (He et al., 2024; Zhou et al., 2025) provide full-frame consistency but do not support personalization to novel reference concepts (cases c, e). Viewpoint-aware subject customization methods (Kumari et al., 2024; Yuan et al., 2023) remain subject-centric, leading to inconsistent surroundings across views (case f). These limitations underscore the need for a new approach explicitly designed for multi-view customization.

Directly adopting multi-view generation frameworks, which rely heavily on large-scale training data, is infeasible in the customization setting, where only a few reference images are available. A straightforward baseline applies conventional customization methods (Ruiz et al., 2023; Hu et al., 2021) directly to text-conditioned multi-view backbones (c in Tbl. 1), but this approach cannot preserve subject identity and reduces camera pose control ability. Another naive baseline generates a single customized image, then applies image-conditioned multi-view generation models (f in Tbl. 1), but the inherent ambiguity of a single view leads to inconsistent spatial relationships and degraded fidelity, as illustrated in Fig. 1.

To address these challenges, we propose *MVCustom*, a diffusion-based framework explicitly designed for robust multi-view customization. Our method separates training and inference stages to effectively

---

*Corresponding author.

handle limited data and ensure geometric consistency across diverse prompts. In the training stage, we leverage pose-conditioned transformer blocks (Kumari et al., 2024). However, a key change is using the video diffusion backbone enhanced with dense spatio-temporal attention to transfer temporal coherence into holistic-frames consistency, ensuring spatial coherence of both the subject and their surroundings across views. At inference, the key challenge is ensuring multi-view geometric consistency for novel prompts, particularly for the subject's surroundings that lack supervision from limited training data. To address this, we introduce two novel inference-stage techniques: *depth-aware feature rendering*, which explicitly enforces geometric consistency using inferred 3D scene geometry, and *consistent-aware latent completion*, which naturally completes previously unseen regions revealed by viewpoint shifts. Extensive comparisons demonstrate that MVCustom is the only approach that effectively integrates accurate multi-view generation and high-fidelity customization.

Our contributions are summarized as follows:

- We propose a novel task, *multi-view customization*, clearly define its requirements, and systematically analyze the limitations of existing methods and tasks.
- We introduce a video diffusion-based backbone enhanced with dense spatio-temporal attention modules, effectively transferring temporal coherence into multi-view consistency.
- To accommodate limited data in customization, we propose two novel inference-stage methods: *depth-aware feature rendering* for explicit geometric consistency, and *consistent-aware latent completion* for consistent and realistic completion of disoccluded regions.

## 2  RELATED WORK

**Conventional text-based customization.**  Customization methods generate images guided by textual prompts while preserving identities from reference images, typically by learning concept-specific embeddings (Gal et al., 2022), fine-tuning models (Ruiz et al., 2023), or applying lightweight adaptations (Hu et al., 2021). Recent approaches further enhance text-image alignment (Alaluf et al., 2023; Li et al., 2024a) and multi-subject control (Kumari et al., 2023; Kwon & Ye, 2024). However, these methods typically lack explicit control over viewpoint. Some works achieve pose-variant compositions (Li et al., 2024b; Song et al., 2024), but do not support explicit camera pose control. Methods like CustomDiffusion360 (Kumari et al., 2024) and CustomNet (Yuan et al., 2023) incorporate viewpoint control yet remain predominantly subject-centric, neglecting to coherently represent their surroundings. In contrast, our proposed *MVCustom* explicitly ensures robust spatial coherence for both customized subjects and surroundings across diverse viewpoints.

**Multi-view generation.**  Multi-view generation models (Zhao et al., 2025; Tang et al., 2024; Alper et al., 2025; Shin et al., 2023) focus on synthesizing consistent multiple views. However, these models typically require large datasets to learn 3D geometry and inpaint newly visible regions, making them unsuitable for customization with only a few reference images. An alternative approach may involve applying conventional customization methods directly onto multi-view generation backbones. Nevertheless, text-conditioned multi-view generation models (Höllein et al., 2024; Shi et al., 2023; Tang et al., 2023; Huang et al., 2024) are limited by the scarcity of paired text and multi-view data, leading to poor adaptability to diverse textual prompts. Another related approach utilizes multi-view diffusion models (Long et al., 2024) for novel-view synthesis from a single reference image, enabling subject-aware editing in multi-view settings (Liu et al., 2024). However, these methods primarily focus only subject editing. In contrast, our *MVCustom* framework explicitly addresses these challenges, combining effective 3D geometry learning with explicit inference-time geometric constraints, enabling robust multi-view consistency and precise alignment with diverse textual prompts.

## 3  METHODOLOGY

In this section, we first introduce our multi-view customization task, explicitly incorporating camera viewpoint control (Sec. 3.1). Next, we describe pose-conditioned transformer blocks to reflect camera poses into the customized subject (Sec. 3.2). Then, we introduce our video diffusion backbone designed for large viewpoint changes (Sec. 3.3). Finally, we present our core contributions — *depth-*

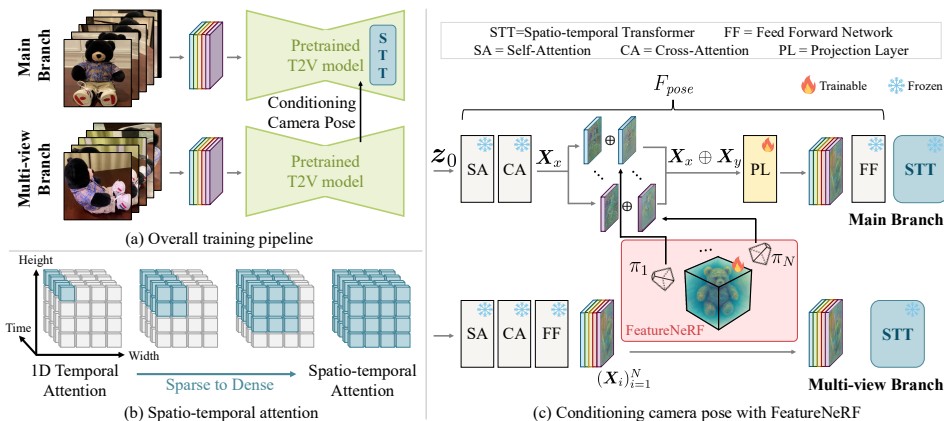

Figure 2: **Overview.** (a) The overall training pipeline, depicting how camera pose conditioning operates with two branches, the main and multi-view. (b) Visualization of our progressive attention mechanism. We gradually broaden the spatial attention field, enhancing geometric consistency. (c) The detailed illustration of the pose-conditioned transformer block. FeatureNeRF and a projection layer are trained to produce a feature map, obtained by concatenating the main-branch and multi-view feature map.

*aware feature rendering* and *consistent-aware latent completion* — to ensure multi-view consistency not only of the customized subject but also their surroundings under novel textual prompts (Sec. 3.4).

## 3.1 PROBLEM DEFINITION

We define *multi-view customization* as an extension of traditional customization that incorporates explicit control over camera viewpoints. Traditional customization aims to model the conditional distribution $p(\boldsymbol{x} \mid \boldsymbol{Y}', \boldsymbol{c})$, where $\boldsymbol{c}$ is a textual prompt describing a novel concept and $\boldsymbol{Y}' = \{\boldsymbol{y}'_i\}_{i=1}^N$ are reference images. A common approach is textual inversion (Gal et al., 2022), which introduces a learnable embedding vector $\boldsymbol{v}$ that replaces part of the text prompt $\boldsymbol{c}(\boldsymbol{v})$. The embedding is learned by minimizing the denoising objective, $\boldsymbol{v}^* = \arg\min_{\boldsymbol{v}} \mathbb{E}_{\boldsymbol{x}, \epsilon \sim \mathcal{N}(0,1), t} \left[\|\epsilon - \epsilon_\theta(\boldsymbol{x}_t; \boldsymbol{c}(\boldsymbol{v}), t)\|_2^2\right]$, where $t$ denotes the diffusion timestep.

In multi-view customization, each reference image is paired with its camera pose, $\boldsymbol{Y} = \{(\boldsymbol{y}_i, \pi_i)\}_{i=1}^N$. The goal is to model the conditional distribution

$$p(\boldsymbol{x}_{0:M} \mid \boldsymbol{Y}, \boldsymbol{c}, \{\phi_m\}_{m=0}^M), \tag{1}$$

where $\boldsymbol{x}_{0:M} = \{\boldsymbol{x}_m\}_{m=0}^M$ denotes a set of generated images under target camera poses $\{\phi_m\}$. For brevity, we denote the set of multi-view outputs as $\boldsymbol{x}$ in the following sections. This formulation enables explicit camera pose control in addition to identity preservation and text alignment, thereby enhancing controllability, consistency, and realism of the generated results.

## 3.2 CONDITIONING CAMERA POSE IN DIFFUSION MODELS

To effectively learn the subject's geometry from reference data, we adopt the pose-conditioned transformer block from CustomDiffusion360 (Kumari et al., 2024), replacing the original spatial transformer in the diffusion models. The transformer block is defined as $F_{pose}(\boldsymbol{z}_0, \{(\boldsymbol{z}_i, \pi_i)\}_{i=1}^N, \boldsymbol{c}, \phi)$, where $\boldsymbol{z}_0$ is the main-branch feature map and $\{(\boldsymbol{z}_i, \pi_i)\}$ are reference features with corresponding poses.

The two branches play complementary roles:

- **Main branch.** Generates target-view features for decoding into the final image. Its feature map is refined via self-attention $s$ and cross-attention $g$ modules conditioned on $\boldsymbol{c}$: $\boldsymbol{X}_x := g(s(\boldsymbol{z}_0), \boldsymbol{c})$.
- **Multi-view branch.** Aggregates reference-view features $\{\boldsymbol{X}_i\}$, computed as $\boldsymbol{X}_i := f(g(s(\boldsymbol{z}_i), \boldsymbol{c}))$. FeatureNeRF synthesizes a pose-aligned feature map $\boldsymbol{X}_y$ by combining $\{\boldsymbol{X}_i\}$

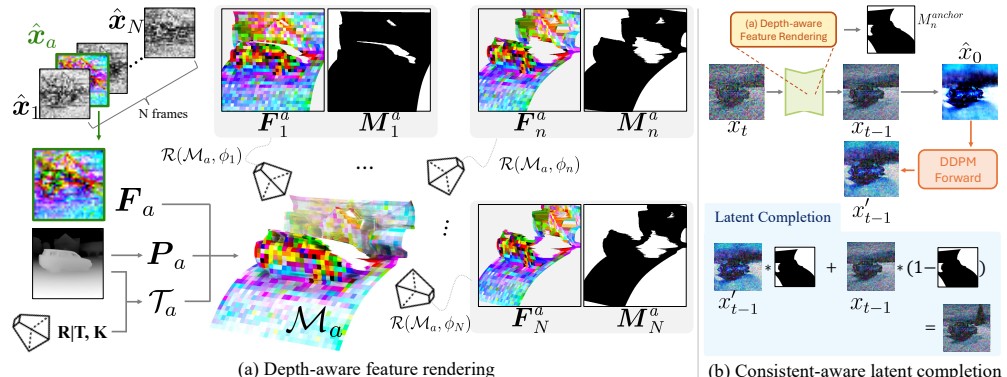

Figure 3: (a) Anchor feature mesh $\mathcal{M}_a$, consists of a texture $\boldsymbol{F}_a$, vertices $\mathbf{P}_a$, and triangles $\mathcal{T}_a$, is constructed using the feature and depth maps, and camera pose of the anchor frame. The $\mathcal{M}_a$ is used to render the projected feature maps for the other camera poses. (b) Completion via latent perturbation for new visible areas.

with camera poses $\{\pi_i\}$ via epipolar geometry (Yu et al., 2021) and volume rendering (Mildenhall et al., 2021):

$$\boldsymbol{X}_y := \text{FeatureNeRF}(\{(\boldsymbol{X}_i, \pi_i)\}_{i=1}^N, \boldsymbol{c}, \phi).$$

These feature maps are concatenated and projected into the backbone's feature space, as shown in Fig. 2a.

### 3.3 BACKBONE FOR DYNAMIC VIEW CHANGE

A pose-conditioned transformer block $F_{pose}$ generally produces consistent multi-view images about the subject, but novel surroundings or clothings are often become inconsistent across views. To address this, we repurpose video generation into multi-view generation based on AnimateDiff (Guo et al., 2023), inherently suited for handling viewpoint transitions. Our video denoising model $D_\theta$ is defined as:

$$D_\theta : (\tilde{\boldsymbol{x}}_{1:N}; \boldsymbol{Y}, \boldsymbol{c}, \phi_{1:N}) \mapsto \hat{\boldsymbol{x}}_{1:N}, \tag{2}$$

mapping noisy inputs $\tilde{\boldsymbol{x}}_{1:N}$ to clean frames $\hat{\boldsymbol{x}}_{1:N}$, conditioned on camera poses $\phi_{1:N}$.

AnimateDiff's 1D temporal attention limits its interactions to identical spatial positions, hindering effective modeling of viewpoint-induced displacements. We extend it with dense 3D spatio-temporal attention (STT) for richer context modeling. To preserve stability and pretrained knowledge, we gradually expand the spatial attention field of STT during training (Fig. 2b). The detailed design choices are discussed in Sec. A.

With this backbone, we fine-tune our customized model by incorporating textual inversion and a pose-conditioned transformer block, optimizing with a standard denoising and additional FeatureNeRF losses (please see Sec. B for the details).

### 3.4 INFERENCE-TIME MULTI-VIEW CONSISTENCY UNDER LIMITED DATA

**Depth-aware feature rendering.** Although our video backbone (Sec. 3.3) produces coherent surroundings, it does not explicitly enforce geometric consistency under camera motion. To address this, we propose *depth-aware feature rendering*, which explicitly imposes geometric constraints conditioned on novel prompts during inference. Unlike previous depth-conditioned multi-view generation methods (Ren et al., 2025; Yu et al., 2024), which rely on large-scale training data, our method effectively addresses the lack of geometric supervision for novel prompt-driven content.

First, the *anchor feature mesh* $\mathcal{M}_a$ is defined using an anchor frame $\hat{\boldsymbol{x}}_a$ selected from $\hat{\boldsymbol{x}}_{1:N}$, denoted as $\mathcal{M}_a = (\boldsymbol{P}_a, \boldsymbol{F}_a, \mathcal{T}_a)$, where the anchor frame's feature map $\boldsymbol{F}_a$ is directly used as texture of

mesh.[1] The vertices $\boldsymbol{P}_a \in \mathcal{R}^{H \times W \times 3}$ are derived from the depth map $D$, estimated by an off-the-shelf depth estimator (Bhat et al., 2023) applied to $\hat{\boldsymbol{x}}_a$. To align the estimated depth $\hat{D}$ with FeatureNeRF's geometric scale, we normalize $\hat{D}$ and shift it by the median depth $d_{\mathrm{med}}$ of the anchor view: $D \leftarrow \mathrm{norm}(\hat{D}) + d_{\mathrm{med}}$. The depth map $D$ is resized to the feature resolution $(H_F, W_F)$ of $\boldsymbol{F}_a$. Using rotation $R \in \mathbb{R}^{3 \times 3}$, translation $T \in \mathbb{R}^3$, and intrinsic matrix $K \in \mathbb{R}^{3 \times 3}$ of the camera parameters associated with $\hat{\boldsymbol{x}}_a$, the 3D points are computed as $\boldsymbol{P} = R(DK^{-1}[u, v, 1]^\top) + T$, where $[u, v]$ denotes a feature-space coordinate. Dense mesh triangles $\mathcal{T}_a$ are defined on the pixel grid using $\hat{D}$, while pruning the regions that become newly visible from other viewpoints, yielding discontinuous mesh boundaries (see Fig. 3a, $\mathcal{M}_a$).

Second, we render $\mathcal{M}_a$ for a given camera pose $\phi_n$, producing the rendered feature map $\boldsymbol{F}_n^a$ and visibility masks $\boldsymbol{M}_n^a$. Notice that the rendering is performed in the feature-space of $\boldsymbol{F}_a$:

$$\boldsymbol{F}_n^a, \boldsymbol{M}_n^a = \mathcal{R}(\mathcal{M}_a, \phi_n), \quad 1 \leq n \leq N, \ n \neq a, \tag{3}$$

where $\mathcal{R}$ denotes a differentiable mesh renderer.

Finally, during the first 35 steps of the 50-step DDIM sampling process, we update each feature map by replacing masked regions with rendered anchor features:

$$\hat{\boldsymbol{F}}_n = \boldsymbol{M}_n^a \odot \boldsymbol{F}_n^a + (1 - \boldsymbol{M}_n^a) \odot \boldsymbol{F}_n, \quad 1 \leq n \leq N, \ n \neq a, \tag{4}$$

then, we substitute the combined feature map $\hat{\boldsymbol{F}}$ for $\boldsymbol{F}$ before the spatial transformer in the second up-block.

**Consistent-aware latent completion.** Regions where $(1 - \boldsymbol{M}_n^a)$ is nonzero correspond to newly visible areas that requires content generation not present in the anchor frame. To address this, we introduce *consistent-aware latent completion*, which leverages stochastic perturbations to synthesize these 'disoccluded' regions (see Fig. 3b). Specifically, given an intermediate noisy latent $x_t$ in the denoising process, we predict an initial latent $x_0$ that is semantically meaningful yet incomplete. We then reintroduce noise into $x_0$ via the forward diffusion process, reverting to the original timestep $t$ and yielding a perturbed latent $x_t'$. The disoccluded regions in the original latent $x_t$ are selectively replaced with those from $x_t'$, enforcing spatial coherence across frames through the temporal consistency of the video backbone. This procedure is iteratively conducted from timestep $T$ down to an early timestep $\tau$ (close to $T$), allowing semantic flexibility and coherent synthesis of novel details in newly exposed regions. Further implementation details, including anchor mesh construction and inference pseudo-code, are provided in Sec. B.

## 4 EXPERIMENT

### 4.1 EXPERIMENTAL SETUP

**Dataset.** We train our video diffusion backbone using a subset (430K samples) of the WebVid10M dataset (Bain et al., 2021). For customization experiments, we use concepts selected from the Common Objects in 3D (CO3Dv2) dataset (Reizenstein et al., 2021), following the setup in CustomDiffusion360 (Kumari et al., 2024). Specifically, we select four categories—car, chair and motorcycle—with three concepts per category. For evaluation, we randomly sample camera trajectories from the CO3Dv2 test set as target camera poses.

**Competitors.** As our task is novel, we compare our proposed method against various applicable baseline approaches: (1) *Custom img + Img-MVgen:* This method generates multi-view images by inputting a single customized image into the image-conditioned multi-view generation model, SEVA (Zhou et al., 2025). The single input image is taken from the first frame of the output produced by our model, conditioned on the target text and camera pose. (2) *Txt-MVgen with DB:* A text-conditioned camera-motion-controllable model, CameraCtrl (He et al., 2024), customized with the conventional DreamBooth-LoRA (Ryu, 2023) approach. (3) *CustomDiffusion360:* An existing object viewpoint-controllable customization method (Kumari et al., 2024). Further comparisons and detailed discussions regarding additional competitors' capabilities and limitations are provided in Sec. C.

---

[1] $\boldsymbol{F}_a$ is the feature map taken immediately before the spatial transformer in the second up-block (Fig. 2c), a feature level previously demonstrated to be effective for diffusion-based feature modification (Go et al., 2024).

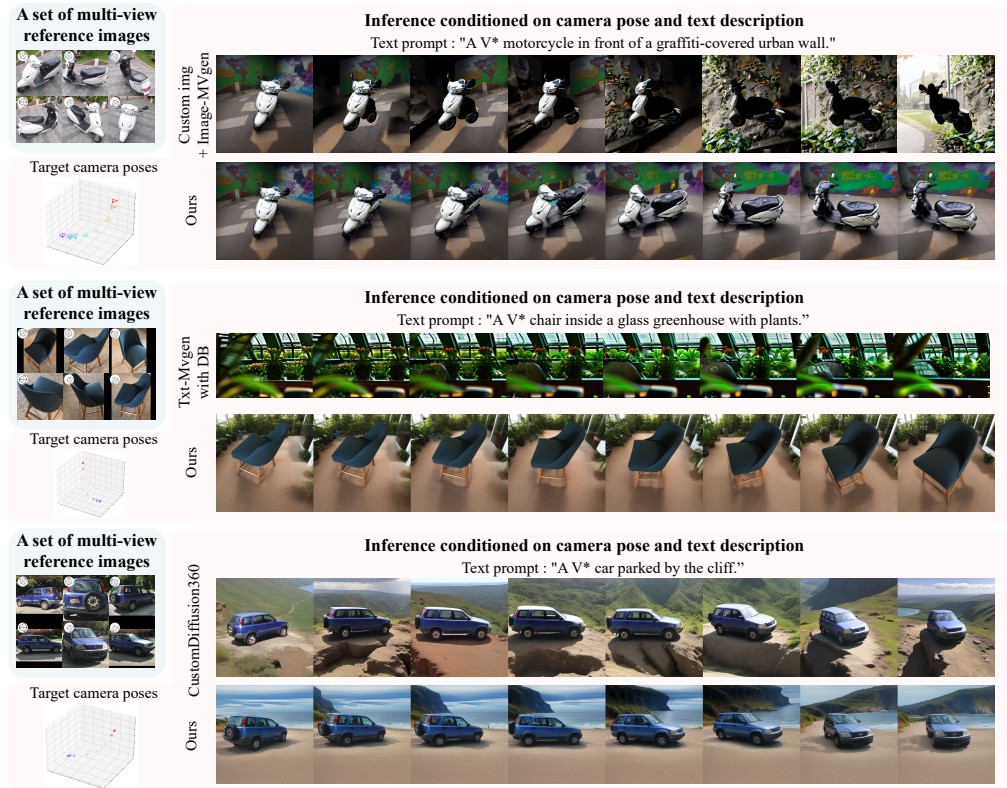

Figure 4: **Qualitative results.** The light blue boxes indicate the multi-view training dataset for the target concept, while the light pink boxes illustrate the inference phase, where results are conditioned on new text and target camera poses.

| Method | MV Generation | | Customization | | Inference Cost | |
|---|---|---|---|---|---|---|
| | Camera Pose Accuracy (↑) | Multi-view Consistency (↓) | Identity Preservation (↓) | Text Alignment (↑) | Time (s) | GPU (GB) |
| Custom Img + Img-MV gen | $0.675 \pm 0.12$ | $0.214 \pm 0.15$ | $0.504 \pm 0.12$ | $0.676 \pm 0.11$ | 96.18 | 6.73 |
| Txt-MV gen with DB | $0.283 \pm 0.25$ | $0.116 \pm 0.09$ | $0.557 \pm 0.12$ | $0.723 \pm 0.10$ | 27.20 | 5.42 |
| CustomDiffusion360 | $0.000 \pm 0.00$ | $0.190 \pm 0.11$ | $0.417 \pm 0.12$ | $0.806 \pm 0.10$ | 74.97 | 4.99 |
| **MVCustom (Ours)** | $0.735 \pm 0.10$ | $0.121 \pm 0.10$ | $0.448 \pm 0.11$ | $0.744 \pm 0.10$ | 130.92 | 19.29 |

Table 2: **Quantitative comparison on multi-view generation, customization, and inference cost.** We highlight the best score in light red and the second-best in yellow.

**Evaluation metrics.** We evaluate our method using four metrics: camera pose accuracy, multi-view consistency, text alignment, and identity preservation. Camera pose accuracy is measured as the average inter-frame relative rotation accuracy (range: [0, 1]), computed via COLMAP (Schonberger & Frahm, 2016). If COLMAP fails to reconstruct camera poses, we assign the minimal accuracy score (0). Multi-view consistency is quantified by visual similarity (Fu et al., 2023) across views, computed over all view pairs. Identity preservation is measured via DreamSim similarity (Fu et al., 2023) between generated outputs and reference images. Text alignment is evaluated using CLIP similarity scores between textual prompts and generated images. Further details and additional evaluations are provided in Sec. C.

## 4.2 RESULTS

As shown quantitatively in Tbl. 2 and qualitatively in Fig. 4, MVCustom is the only approach that simultaneously achieves high multi-view consistency and accurate customization fidelity. More comprehensive video comparisons can be found in the project page.

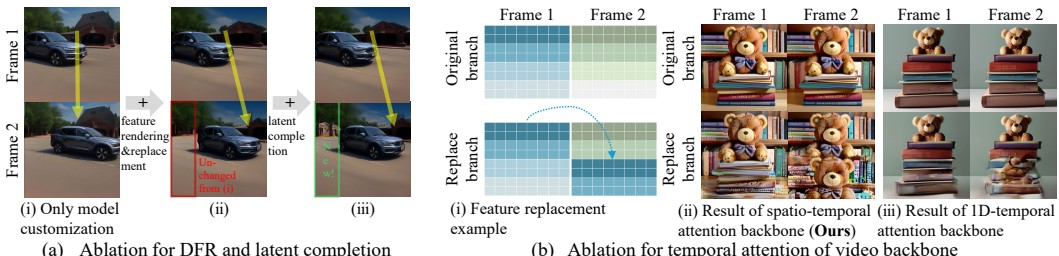

(a)   Ablation for DFR and latent completion          (b)   Ablation for temporal attention of video backbone

Figure 5: **Results of ablation studies.** (a) Stepwise effect of applying depth-aware feature rendering (DFR) and consistent-aware latent completion under x-translation camera pose. (b) Impact of temporal attention on feature replacement. (i) Feature replacement vertically copies the feature map from frame 1 to frame 2. Our method successfully enforces spatial flow, whereas 1D temporal attention fails to capture the intended translation.

**Multi-view consistency with perspective alignment.**    Accurately reflecting target camera poses is crucial for multi-view customization. As shown in Tbl. 2 (camera pose accuracy) and qualitative examples (Fig. 4), MVCustom faithfully generates multi-view images aligned with specified view-points. In contrast, *Txt-MV gen with DB* fails to reflect rotation-aware trajectories despite explicit conditioning, as clearly observed in the chair example of Fig. 4, and confirmed by poor pose accuracy (Tbl. 2). This indicates that the strong camera controllability in Txt-MV generation does not directly translate into multi-view customization through conventional fine-tuning (see Sec. C.3). Similarly, *Img-MV gen* methods rely on a single reference image, limiting subject appearance and geometry, and causing unnatural subject–surrounding relationships in distant views (e.g., the motorcycle in Fig. 4). Although *CustomDiffusion360* maintains subject consistency, arbitrary surroundings across viewpoints yield poor holistic multi-view consistency, leading to COLMAP reconstruction failure and zero pose accuracy (Tbl. 2). By leveraging our video backbone and inference strategies, MVCustom substantially improves holistic multi-view consistency and perspective alignment, outperforming all baselines.

As shown in Tbl. 2, MVCustom requires higher computational resources primarily due to the external depth estimator (increasing GPU memory) and the feature replacement step (increasing inference time), unlike other competitors relying solely on denoising. Nevertheless, explicitly enforcing geometric consistency at inference is critical given the constraint of extremely limited training data. Thus, we argue that our significant improvements in multi-view consistency, geometric accuracy, and customization fidelity clearly justify this computational trade-off.

**ID preservation with text alignment**    The *Custom img + Img-MV gen* baseline fails to preserve subject identity and the textual description of surroundings, particularly as viewpoints move further from the input image (as shown qualitatively in Fig. 4). *Txt-MV gen with DB* also fails to retain the reference subject's appearance and geometry, leading to poor identity preservation. In contrast, both *CustomDiffusion360* and our MVCustom method successfully preserve the reference subject and effectively reflect diverse textual prompts across all views, demonstrating superior customization fidelity.

### 4.3    ABLATION STUDY

**Depth-aware feature rendering & Consistent-aware latent Completion.**    Customization fine-tuning alone yields static surroundings despite varying subject poses (Fig. 5a-i). Our novel depth-aware feature rendering enforces geometric consistency, enabling accurate spatial shifts (e.g., building position) according to camera movements (Fig. 5a-ii). However, newly revealed regions reuse previous content, reducing realism. Thus, we propose latent completion, leveraging the generative power of our diffusion backbone to naturally synthesize previously unseen, context-appropriate details (Fig. 5c). Unlike conventional multi-view methods requiring extensive datasets, our method explicitly addresses data limitations in customization, significantly enhancing multi-view coherence and realism; see Sec. D and Sec. E for additional ablation samples and completion results demonstrating visual diversity.

**Spatio-temporal attention.** We evaluate dense spatio-temporal attention's effectiveness for spatial consistency. As illustrated in Fig. 5b-i, we vertically shift and insert the first frame's features into subsequent frames, expecting clear semantic translations. While original AnimateDiff with 1D temporal attention fails to preserve spatial coherence due to limited pixel interactions (Fig. 5b-ii), our proposed spatio-temporal attention successfully maintains spatial consistency and semantic flow (Fig. 5b-iii). Thus, integrated spatio-temporal attention is crucial for accurately modeling large view displacements and explicitly enforcing spatial constraints, especially when employing feature replacement (Sec. 3.4).

## 5 CONCLUSION

In this work, we introduced the novel task of *multi-view customization*, integrating explicit camera viewpoint control, subject customization, and spatial consistency for both subjects and their surroundings. To address this task, we proposed *MVCustom*, a diffusion-based framework leveraging dense spatio-temporal attention for robust multi-view synthesis. Additionally, we introduced two inference-stage strategies—*depth-aware feature rendering* and *consistent-aware latent completion*—to explicitly enforce geometric consistency and faithfully generate disoccluded regions. Extensive comparisons show that MVCustom is the only approach that effectively integrates accurate multi-view generation and high-fidelity customization. We believe this framework provides a foundation for future work on controllable and customizable multi-view generation.

**Limitations and future work** Our framework currently cannot alter the intrinsic object pose based on text prompts during inference (e.g., changing from sitting to standing). This limitation arises because FeatureNeRF learns a fixed canonical pose from reference images, and its radiance field does not take text prompts as input conditions. Consequently, the object's intrinsic pose remains tied to this canonical representation. Experimentally, we found that injecting the rendered feature map $X_y$ via cross attention conditioned on textual prompts does not overcome this issue. Similar limitations related to intrinsic pose control are noted in prior customization work (Song et al., 2024). Future approaches might involve optimizing a dynamic neural field conditioned on textual prompts built upon a frozen static field from FeatureNeRF, using techniques such as score distillation sampling, or hypernetwork-based methods. We leave these directions for future exploration. However, some text-specified shape variations can be reflected by adjusting the usage ratio of pose-conditioned transformer blocks during inference steps, as detailed in Sec. D.1.

Additionally, another limitation arises from inaccuracies in the depth maps used in our depth-aware feature rendering. When the external depth estimator produces incorrect geometry, especially for reflective or textureless surfaces, our method directly constructs feature meshes using these inaccuracies. This limitation originates from the external depth estimator rather than our framework itself. Similar issues affect other depth-conditioned methods (Yang et al., 2025; Liu et al., 2025; Hou & Chen, 2024) due to their inherent dependence on accurate depth maps. Recent models (Yang et al., 2024;

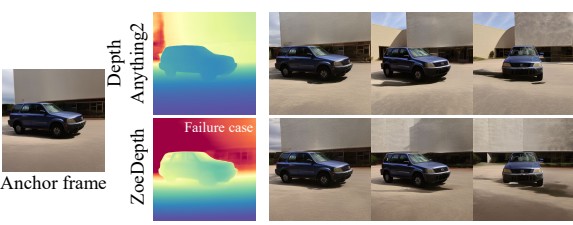

Figure 6: Comparison of background perspective alignment in generated images depending on the quality of estimated depth.

Min et al., 2025) have significantly improved depth estimation accuracy for reflective and textureless surfaces, suggesting potential mitigation of this issue. Fig. 6 demonstrates that accurate depth estimation produces realistic background geometry across multiple views: correctly estimating the depth of a textureless wall ensures the building naturally rotates with the viewpoint change. Conversely, incorrect estimation perceiving the wall as distant background results in unrealistic backgrounds across views. In conclusion, we expect that ongoing advancements in depth estimation techniques will soon overcome this limitation, enabling our framework to produce even more realistic and consistent multi-view results.

**Acknowledgements** This work was supported by IITP grants [RS-2024-00439762, Developing Techniques for Analyzing and Assessing Vulnerabilities, and Tools for Confidentiality Evaluation in

Generative AI Models] and [RS-2020-II201361, Artificial Intelligence Graduate School Program (Yonsei University)] funded by the Korean government (MSIT) .

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

## A    REASON FOR OUR DESIGN CHOICE

In this section, we clarify the rationale behind our architectural design choices.

### A.1    U-NET-BASED DIFFUSION MODEL

We specifically choose a U-Net-based video diffusion model rather than recent DiT-based models, primarily for architectural compatibility with FeatureNeRF (Kumari et al., 2024), which serves as the starting point of our method. DiT models rely on Conv3D-based VAE, merging spatial and temporal dimensions. Consequently, these models cannot guarantee a consistent number of features per frame, crucial for accurate frame-level camera pose conditioning. In contrast, our U-Net-based model explicitly maintains per-frame feature maps, ensuring effective camera pose conditioning.

Among available U-Net-based text-to-video models, we build upon AnimateDiff (Guo et al., 2023) due to its state-of-the-art video generation capability and compatibility with diverse stylizations such as DreamBooth (Ruiz et al., 2023) and LoRA (Hu et al., 2021). As illustrated in figure A1, incorporating various DreamBooth models significantly enhances style controllability without altering the customized object's identity. For photo-realistic rendering, we integrate our customization model with RealisticVision[1] for all experiments.

**Justification for U-Net over DiT in multi-view customization task**    While a U-Net-based backbone may slightly compromise visual quality compared to transformer-based architectures, our primary goal is accurate geometry learning and effective frame-level control under extremely limited data conditions. DiT-based video models typically require training an additional conditioning network for precise frame-level camera pose control (Bai et al., 2025; Bahmani et al., 2025; Bai et al., 2024). Such approaches rely heavily on large-scale datasets; for example, RelCamMaster uses 136K videos and AC3D employs 65M videos. In contrast, our multi-view customization scenario provides only a single video sample (approximately 50 frames), insufficient to reliably train additional networks without severe overfitting and loss of controllability. Recognizing the fundamental limitations of DiT-based architectures under these constrained conditions, we adopted a U-Net architecture that maintains per-frame latent features. This choice enables us to explicitly impose geometry constraints during inference using Feature Mesh Rendering, thus achieving robust viewpoint controllability and geometric consistency.

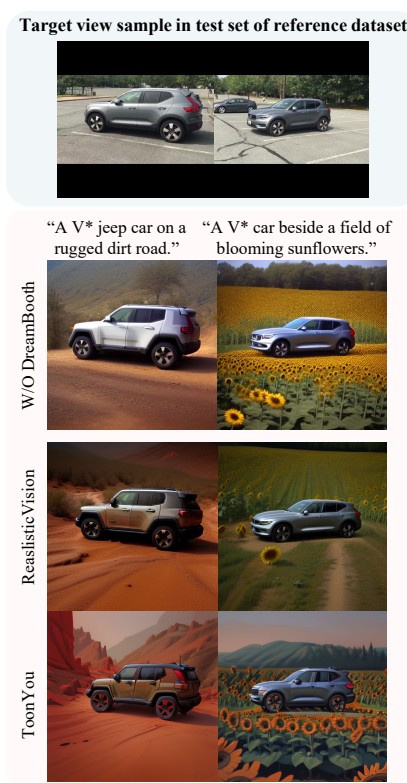

**Target view sample in test set of reference dataset**

"A V* jeep car on a rugged dirt road."    "A V* car beside a field of blooming sunflowers."

W/O DreamBooth

RealisticVision

ToonYou

Figure A1: **Results with different DreamBooth models.** Since our method keeps spatial transformer layers of the video backbone architecture frozen, we can flexibly apply various publicly available DreamBooth checkpoints. The figure shows images generated using two different checkpoints: RealisticVision[1] and ToonYou[2].

### A.2    NUMBER OF FEATURENERF MODULES.

The number of FeatureNeRF modules has a trade-off between accurately preserving the identity of the reference object and effectively reflecting new textual descriptions. Increasing the number of transformer blocks with FeatureNeRF better preserves identity, as these modules emphasize the reference object's details. However, this approach makes the model less responsive to novel textual descriptions during inference, because the projection layers, after the concatenation of the multi-view

---

[1] https://civitai.com/models/4201?modelVersionId=130072
[2] https://civitai.com/models/30240/toonyou

and main branches, are biased towards the reference branch rather than the main branch which directly processes new text conditions. Conversely, decreasing the proportion of FeatureNeRF modules enhances the model's ability to reflect diverse textual prompts, but weakens identity preservation due to the reduced influence of the rendered radiance field from the reference object. Our choice of employing FeatureNeRF in 7 out of 16 transformer blocks represents a balanced compromise, ensuring both faithful identity preservation and robust adaptability to new textual inputs.

# B  IMPLEMENTATION DETAILS

## B.1  VIDEO BACKBONE

We adopt the 1D temporal attention model from AnimateDiff (Guo et al., 2023) as our starting point. To integrate this model effectively into our customization framework, we first reduce the number of generated frames. Although AnimateDiff originally generates 16-frame videos, simultaneous generation of 16 frames in both our main and multi-view branches leads to GPU memory constraints. Therefore, we fine-tune the backbone to generate 8 frames, preserving its original 1D temporal attention structure. All quantitative evaluations in this study utilize this 8-frame version.

During the initial fine-tuning, only the temporal transformer modules are trained using a denoising loss. Training is conducted for 100 steps with the Adam optimizer and a learning rate of $1 \times 10^{-4}$.

Subsequently, as detailed in Section 3.3, we transition progressively from sparse temporal attention to dense spatio-temporal attention. Again, fine-tuning focuses exclusively on the temporal transformer modules. The resolution of attention feature maps gradually increases from $2^0$ to $2^6$, doubling every 10k training steps. This interval remains consistent even for resolutions below 64, enabling faster formation of dense spatio-temporal attention at lower resolutions. Following AnimateDiff's original practice, domain adapters are utilized only during training and removed thereafter.

We train our model using a subset of 430k videos from WebVid10M (Bain et al., 2021), selected for dynamic scores above 80, at a resolution of 512 pixels. The training employs the Adam optimizer with a learning rate of $1 \times 10^{-4}$ and the DDPM scheduler. The entire training process takes approximately one week on four NVIDIA A6000 GPUs with a per-GPU batch size of 2.

**Fine-tuning for model customization.**  We perform model customization on top of our video backbone equipped with the proposed spatio-temporal attention, generating 8-frame videos at a resolution of 512 pixels. During customization, both the main and multi-view branches generate 8-frame videos. The dataset for each concept is sourced from CustomDiffusion360 (Kumari et al., 2024). The trainable parameters include the concept-specific text embeddings optimized via textual inversion (Gal et al., 2022), as well as the NeRF MLP and projection layers of FeatureNeRF.

Following CustomDiffusion360, at the end of customization training, each FeatureNeRF stores intermediate feature maps $(\boldsymbol{X}_i)_{i=1}^{\#\text{ReferenceDataset}}$ from the training dataset. While CustomDiffusion360 stores these intermediate feature maps at random timesteps, our method specifically stores them at timestep 10 (close to the clean-image timestep) out of the total 1000 timesteps.

We adopt the loss weighting scheme from CustomDiffusion360 for both FeatureNeRF and textual inversion, and training is performed using the DDPM scheduler. Fine-tuning each concept takes approximately one day on a single NVIDIA A6000 GPU, using the Adam8bit optimizer with a learning rate of $1 \times 10^{-4}$.

## B.2  INFERENCE STAGE: DEPTH-AWARE FEATURE RENDERING.

We describe the detailed procedure for constructing the anchor feature mesh used in our feature rendering method.

The texture $\mathbf{C}$ of the mesh is directly obtained from the anchor frame's feature map $F_a$.

The 3D vertices $\mathbf{P} = (X, Y, Z)^\top$ are generated based on depth $D$, estimated from the anchor frame $\hat{x}_a$ using an off-the-shelf depth estimator (Bhat et al., 2023). To align the estimated depth $\hat{D}$ with FeatureNeRF's learned geometry, we scale it using the median depth $d_{\text{med}}$ computed from the central ray of the anchor frame, as $D = \hat{D}/|\hat{D}| + d_{\text{med}}$. Here, we initially use the median depth from the

first FeatureNeRF view. If $d_{\text{med}}$ is inaccurate, the position of the rendered object may not align with the object generated by FeatureNeRF for target camera poses, negatively impacting the perspective alignment of the background harmonized around the FeatureNeRF-rendered object. To resolve this, we conduct a grid search within a $\pm 40\%$ range around $d_{\text{med}}$, selecting the optimal $d'_{\text{med}}$ that minimizes the error between the object region of frames generated without feature rendering and the RGB mesh produced using $D = \hat{D}/|\hat{D}| + d'_{\text{med}}$. The foreground object region is defined by the alpha mask rendered by FeatureNeRF.

The depth $D$ is resized to match the feature resolution $(H_F, W_F)$. Using rotation $R \in \mathbb{R}^{3 \times 3}$, translation $T \in \mathbb{R}^3$, and intrinsic matrix $K \in \mathbb{R}^{3 \times 3}$ from the anchor frame, we define the 3D points as $\mathbf{P} = R\mathbf{P}_c + T$, where $\mathbf{P}_c = DK^{-1}[u, v, 1]^\top$, and $[u, v]$ are image coordinates at feature resolution $(H_F, W_F)$.

To define triangles $\mathcal{T}$, we first create a regular grid of triangles $\mathcal{T}_{\text{raw}}$ based on $\hat{D}$. We then exclude triangles corresponding to depth discontinuities, which represent regions not visible from the anchor view but potentially visible from other viewpoints due to occlusions. Triangles are validated using:

$$V(t) = \begin{cases} 1, & \min_{(i,j) \in t} \sqrt{\left(\frac{\partial D(i,j)}{\partial x}\right)^2 + \left(\frac{\partial D(i,j)}{\partial y}\right)^2} > \zeta, \\ 0, & \text{otherwise} \end{cases}$$

where $\zeta = 0.05$ is a threshold for significant depth variations. The final triangle set is then:

$$\mathcal{T} = \{t \in \mathcal{T}_{\text{raw}} \mid V(t) = 1\}.$$

During mesh rendering $\mathcal{R}(\mathcal{M}, \phi_n)$, lighting and shading effects are not considered.

This feature replacement is performed at the second spatial transformer block in the U-Net decoder, specifically after the ResNet module and before feeding the feature map into the subsequent feed-forward network.

### B.3 INFERENCE STAGE: CONSISTENT-AWARE LATENT COMPLETION.

For the primary inference from pure noise $x_T$ to clean latent $x_0$, we use the deterministic DDIM scheduler (ODE). However, for creating perturbed latent $x'_T$ during latent completion, we adopt the stochastic DDPM forward process (SDE manner). The timestep $\tau$ for latent completion is set at step 15 out of the total 50 inference steps.

We include pseudo-code in Algorithm 1 to illustrate the sequence of operations for depth-aware feature rendering and consistent-aware latent completion.

---

**Algorithm 1** Depth-aware Feature Rendering and Consistent-aware Latent Completion.

---

**Require :** RGB frames $I_{1:N}$, feature maps $F_{1:N}$, camera poses $\{\phi_n\}_{1:N}$, total diffusion timesteps $t_{\text{tot}} = 50$, replacement diffusion timesteps $t_{\text{rep}} = 35$
**Notation :** For any quantity with subscript $n$ (e.g. $I_n$, $F_n$, $\phi_n$), the index $n \in \{1, \dots, N\}$ refers to the $n$-th frame. $n_a$ refers to selected anchor frame.

---

**PART 1: Prepare Anchor Mesh**
**function** PREPAREANCHORMESH($\hat{D}, F_{n_a}, K_{n_a}, T_{n_a}, R_{n_a}$)
    $d_{\text{med}} \leftarrow \text{MEDIANDEPTH}(\hat{D})$
    $D \leftarrow \hat{D}/|\hat{D}| + d_{\text{med}}; D \leftarrow \text{RESIZE}(D, (H_F, W_F))$
    $P_c[u, v] \leftarrow D[u, v] \cdot K_{n_a}^{-1} \begin{pmatrix} u \\ v \\ 1 \end{pmatrix}$
    $P[u, v] \leftarrow R_{n_a} P_c[u, v] + T_{n_a}$
    $\mathcal{T}_{\text{raw}} \leftarrow \text{GRIDTRIANGLES}(D)$
    $\mathcal{T} \leftarrow \{t \in \mathcal{T}_{\text{raw}} \mid \min \nabla D(t) > \zeta\}$
    $\mathcal{M} \leftarrow \text{MESH}(P, \mathcal{T}, F_{n_a})$
    **return** $\mathcal{M}$

---

**PART 2: Inference stage with Depth-aware Feature Rendering & Replacement and Consistent-aware Latent Completion**
**for** $t = 1$ **to** $t_{\text{tot}}$ **do**
    **if** $t \leq t_{\text{rep}}$ **then**
        $n_a \leftarrow \text{CHOOSEANCHORFRAME}()$
        $\hat{D} \leftarrow \text{DEPTHESTIMATOR}(I_{n_a})$
        $(K_{n_a}, T_{n_a}, R_{n_a}) \leftarrow \phi_{n_a}$
        **for all** $n \in \{1, \dots, N\} \setminus \{n_a\}$ **do**
            $\mathcal{M} \leftarrow \text{PREPAREANCHORMESH}(\hat{D}, F_{n_a}, K_{n_a}, T_{n_a}, R_{n_a})$
            $(F_n^{\text{anchor}}, M_n^{\text{anchor}}) \leftarrow \text{RENDER}(\mathcal{M}, \phi_n)$       ▷ Feature rendering
            $F_n \leftarrow M_n^{\text{anchor}} \odot F_n^{\text{anchor}} + (1 - M_n^{\text{anchor}}) \odot F_n$   ▷ Feature replacement
        $x_t \leftarrow \text{ENCODELATENT}(F_n)$
        $x_0 \leftarrow \text{PREDICTCLEANLATENT}(x_t)$
        $x_t' \leftarrow \text{DIFFUSIONFORWARDPROCESS}(x_0)$
        $x_{new} = x_t' \odot (1 - M_n^{\text{anchor}}) + x_t \odot M_n^{\text{anchor}}$    ▷ Completion for disocclusion
        $F_n \leftarrow \text{DECODELATENT}(x_{new})$

---

## C    DETAILS OF EVALUATION

### C.1   COMPETITORS

We provide detailed explanations regarding the evaluation setups and limitations of our main competitors, namely *Brute Force (Customized single image + SEVA)* and CustomDiffusion360. Additionally, we include an evaluation of CustomNet, another relevant method applicable to multi-view customization.

**Custom Img + Img-MV gen.** We consider a straightforward baseline, named *Custom Img + Img-MV gen*, which involves feeding a single customized image reflecting a text description into an image-conditioned multi-view diffusion model. We specifically adopt SEVA (Zhou et al., 2025), the state-of-the-art image-conditioned multi-view diffusion model, for this baseline.

Although SEVA can accept multiple image inputs, achieving multi-view consistency among customized images that reflect novel textual descriptions remains challenging in multi-view customization tasks. Thus, this baseline uses only a single customized image as input to SEVA. The single customized image used as input is taken from the first frame generated by our method.

To evaluate Brute Force under the best conditions, we use the official target views provided by the SEVA implementation. Specifically, we select an "orbit" trajectory from the test set for camera pose

evaluation, choosing "move-left" for positive x-translation and "move-up" for positive y-translation. We generate a total of 34 frames from SEVA, from which 8 frames (including the input image as the first frame) are sampled for evaluation.

**Txt-MV gen with DB.** We trained a DreamBooth-LoRA (Ryu, 2023) on Stable Diffusion using 50 reference images for 2000 steps, and then integrated the customized LoRA into CameraCtrl (He et al., 2024).

**CustomDiffusion360.** Since CustomDiffusion360 (Kumari et al., 2024) is built on a text-to-image model, the generated semantics differ significantly even with slight variations in camera pose, despite using identical noise and text prompts. Although the surrounding semantics may appear similar across different views, this similarity mainly results from partial overfitting to the prior preservation dataset. Thus, while CustomDiffusion360 provides effective object pose controllability and customization capability, it does not explicitly address multi-view consistency. We evaluate CustomDiffusion360 using the official checkpoint provided in the original repository.

## C.2 DETAILS OF THE QUANTITATIVE EVALUATION PROTOCOL

We evaluate our method on 14 concepts, each with 16 text prompts. We use the same set of evaluation prompts provided in the supplementary material of CustomDiffusion360. To ensure a fair comparison, all models share this common set of text prompts.

For each prompt, our method generates 8 images from different viewpoints. The target camera poses are randomly sampled from trajectories provided in the test set of the reference dataset.

**MV-Consistency.** To quantify the multi-view (MV) consistency of generated images across viewpoints, we measure visual similarity using DreamSim. Similarly, we conduct additional analyses in table A1 using image-based similarity metrics computed by CLIP ViT-B/32 (Radford et al., 2021) and DINO ViT-S/16 (Caron et al., 2021). We compute these metrics across all pairwise combinations of images generated from the same concept and textual prompt. For DreamSim, we follow the official implementation, where lower values indicate higher perceptual similarity. For CLIP and DINO similarities, we extract features from generated images, with higher scores indicating better similarity. Our method consistently achieves the highest scores across all three metrics, demonstrating strong preservation of subject consistency in multi-view images.

Additionally, we evaluate geometric alignment using the Met3R metric (Asim et al., 2025), which quantifies the consistency of 3D structures and semantics between pairs of generated images from different viewpoints. Following the original Met3R protocol, we compute pairwise scores for all adjacent frame pairs and average them to obtain the final MV-consistency score. Lower Met3R scores indicate higher consistency. However, Met3R does not explicitly evaluate alignment to the target camera poses, as evidenced by favorable evaluations even when camera poses are completely disregarded, such as in *Txt-MV gen with DB*.

**Camera Pose Accuracy.** We evaluate camera controllability using Camera Pose Accuracy (CPA), normalized between 0 and 1. We exclusively focus on rotations since translation scales are inconsistent across methods: ours and CustomDiffusion360 (Kumari et al., 2024) use normalized poses, while CameraCtrl (He et al., 2024) and SEVA (Zhou et al., 2025) do not. Direct translation comparisons would therefore confuse controllability with scale mismatches.

Given a target camera pose sequence $R_{\text{gen}}^j$ and estimated poses $R_{\text{est}}^j$ obtained from COLMAP on the generated video, the angular deviation for each frame is defined as:

$$\theta^j = \arccos\left(\frac{\text{tr}(R_{\text{est}}^j R_{\text{gen}}^{j\top}) - 1}{2}\right), \quad \theta^j \in [0, \pi]. \tag{5}$$

This angular error is converted into a per-frame accuracy:

$$a^j = 1 - \frac{\theta^j}{\pi}, \quad a^j \in [0, 1], \tag{6}$$

| Method | Met3R ($\downarrow$) | CLIP image similarity ($\uparrow$) | DINO image similarity ($\uparrow$) |
|---|---|---|---|
| Custom Img + Img-MV gen | $0.252 \pm 0.078$ | $0.877 \pm 0.067$ | $0.759 \pm 0.147$ |
| Txt-MV gen with DB | $0.216 \pm 0.107$ | $0.927 \pm 0.044$ | $0.868 \pm 0.096$ |
| CustomDiffusion360 | $0.400 \pm 0.085$ | $0.890 \pm 0.056$ | $0.802 \pm 0.095$ |
| **MVCustom (Ours)** | $0.265 \pm 0.154$ | $0.933 \pm 0.048$ | $0.868 \pm 0.097$ |

Table A1: **Additional quantitative evaluation of multi-view consistency.** Our method achieves the highest multi-view consistency across all three image similarity metrics, demonstrating that the generated images exhibit strong alignment and similarity with each other across different viewpoints.

where $a^j = 1$ indicates perfect alignment and $a^j = 0$ corresponds to a rotation difference of $180°$.

The *sample-level CPA* for a video with $N$ frames is computed as:

$$\text{CPA}_{\text{sample}} = \frac{1}{N} \sum_{j=1}^{N} a^j. \tag{7}$$

The final *dataset-level CPA* averages all $M$ evaluation samples:

$$\text{CPA}_{\text{final}} = \frac{1}{M} \sum_{i=1}^{M} \text{CPA}_{\text{sample}}^{(i)}. \tag{8}$$

We adopt the following failure handling strategy for robustness and fairness:

- **Full reconstruction failure:** If COLMAP fails entirely to reconstruct a trajectory due to unsuccessful feature matching, we assign $\text{CPA}_{\text{sample}} = 0$.
- **Partial pose failure:** If COLMAP reconstructs partially but fails for certain frames, we set those frames' accuracy to $a^j = 0$, which contributes to the average CPA.

This ensures that the reported CPA accurately reflects controllable camera trajectories and penalizes both sequence-level and frame-level estimation failures.

**Reference image fidelity.** To evaluate how well the generated images depict the concepts present in the reference images, we measure the perceptual similarity using DreamSim (Fu et al., 2023). Since DreamSim effectively captures semantic content, using a subset of reference images yields results comparable to using the full set. Therefore, for efficiency, we construct the reference set by randomly sampling reference images for each text prompt, matching the number of generated images. We then compute DreamSim between each generated image and all sampled reference images, and report the average similarity across all concepts and text prompts.

**Text alignment.** CLIP text-image similarity is computed between each generated image and its corresponding prompt using the CLIP ViT-B/32 model (Radford et al., 2021). We compute the similarity between each generated image and its corresponding text prompt and report the average score as the final result. Higher similarity scores indicate better text alignment of the generated images.

## C.3 Limitations of standard customization on camera-controllable models.

Applying standard image customization methods (e.g., DreamBooth-LoRA) to text-conditioned camera pose controllable models (e.g., CameraCtrl) significantly reduces camera pose controllability.

These results show that simply applying image customization to a text-conditioned multi-view generation model does not achieve multi-view customization. Therefore, a new framework specifically designed for the goal of multi-view customization is necessary.

## D Further Ablation on Inference Strategies

This section provides additional analysis of our inference strategies: depth-aware feature rendering and consistent-aware latent completion. With only model customization (fine-tuning), the target

| Method | Rotation Error (↓) | Translation Error (↓) |
|---|---|---|
| CameraCtrl | **15.660** | **4.385** |
| CameraCtrl + DB-LoRA | 16.500 | 4.608 |

Table A2: **Effect of naive customization on CameraCtrl.** Evaluation follows the protocol of CameraCtrl: rotation error is measured in degrees, and target poses are randomly sampled from its public trajectory set.

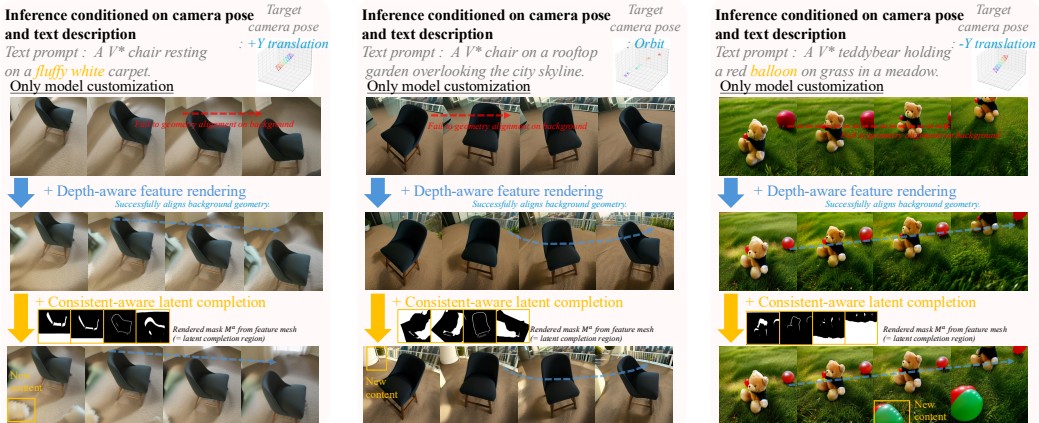

Figure A2: **Results on ablation study.**

camera trajectory aligns solely with the object, as shown in the first row of Fig. A2, while the surroundings remain static or unrelated to the camera pose. Adding depth-aware feature rendering ensures that the surroundings accurately reflect the target camera trajectory, as illustrated in the middle row of Fig. A2. This enhancement significantly improves perspective alignment, which is quantitatively supported by the substantial improvement in multi-view generation scores in Tbl. A3. However, disoccluded regions retain previous content, lacking diversity. Introducing consistent-aware latent completion further improves realism by adding contextually appropriate new content into disoccluded regions, as demonstrated in the bottom row of Fig. A2. While this addition slightly decreases the multi-view visual consistency metric due to reduced semantic similarity across frames, it does not imply visual inconsistency. The improved COLMAP reconstruction points and pose accuracy validate this.

## D.1 ANALYSIS OF OBJECT SHAPE VARIANTS FROM TEXT PROMPT

We analyze the extent of shape variation achievable by adjusting the usage ratio of trained pose-conditioned transformer blocks (containing FeatureNeRF) versus the backbone's original transformer blocks during inference denoising steps. Specifically, we investigate the following scenarios:

- **Case (a) – 100% pose conditioning:** Trained pose-conditioned transformer blocks are applied throughout all inference steps. This accurately maintains viewpoint alignment, preserves the reference object's shape, and partially reflects shape variations specified by the text prompt.

- **Case (b) – 75% pose conditioning:** Trained pose-conditioned transformer blocks are applied only during the initial 75% of the inference steps, with the backbone's original transformer blocks used in the remaining steps. This setting allows greater reflection of text-specified shape variations while slightly compromising reference shape consistency.

- **Case (c) – 50% pose conditioning:** Trained pose-conditioned transformer blocks are applied in the initial 50% of inference steps, and the remaining steps use the backbone's original transformer blocks. This reflects detailed shape variations from text prompts while partially preserving key reference features (e.g., color or door shape), though viewpoint alignment begins to degrade.

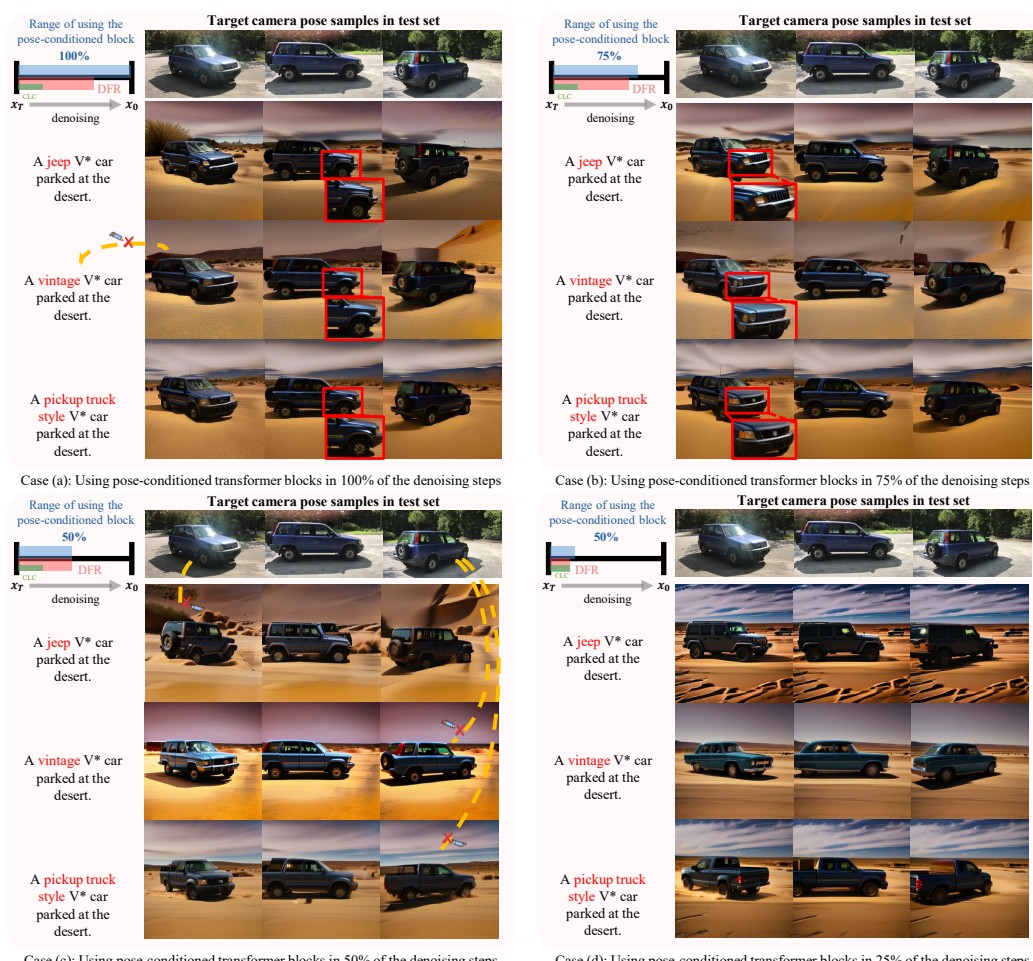

Figure A3: **Object shape variants achieved by varying the ratio of pose-conditioned transformer blocks.** The ratio indicates the proportion of inference denoising steps that utilize the trained pose-conditioned transformer blocks. For example, 100% means that trained pose-conditioned transformer blocks are used throughout all inference steps, whereas 75% means they are used only for the initial 75% of steps, with the remaining steps employing the backbone's original transformer blocks.

| Method | Multi-view Generation | | | Customization | |
|---|---|---|---|---|---|
| | # of Colmap recon (↑) | PoseAcc (↑) | Multi-view consistency (↓) | Identity Preservation (↓) | Text Alignment (↑) |
| Only customization | 36.13 ± 19.87 | 0.543 ± 0.179 | 0.095 ± 0.067 | 0.355 ± 0.076 | **0.682 ± 0.074** |
| + Depth-aware feature rendering | 43.38 ± 15.98 | 0.768 ± 0.153 | **0.090 ± 0.065** | **0.347 ± 0.068** | 0.679 ± 0.073 |
| + Consistent-aware latent completion | **45.38 ± 26.39** | **0.771 ± 0.142** | 0.113 ± 0.081 | 0.384 ± 0.086 | 0.681 ± 0.066 |

Table A3: **Quantitative evaluation of inference strategies.** "# of COLMAP recon" indicate the average number of reconstructed points from multi-view images with target camera poses by COLMAP. Best results are highlighted in bold; second-best in italic. Evaluations conducted on rotation-aware camera trajectory and translation trajectory.

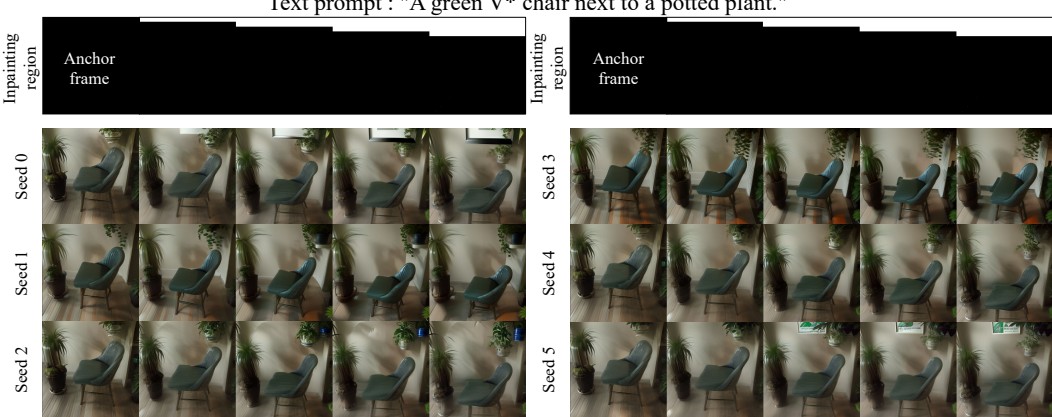

Figure A4: **Diversity of consistent-aware latent completion.** The white regions in the top row denote completion areas. The variations across seeds reflect the diversity induced by noise randomness.

- **Case (d) – 25% pose conditioning:** Trained pose-conditioned transformer blocks are applied only during the initial 25% of inference steps, after which the backbone's original transformer blocks are utilized. This configuration does not retain reference object details or viewpoint consistency, accurately reflecting only the general object type and details described by the text. Hence, it does not constitute meaningful reference customization.

These shape variations specifically affect only the customized object, leaving the surroundings unaffected.

## E  DIVERSITY OF LATENT COMPLETION

In our method, after constructing the anchor feature mesh from an anchor frame, we employ latent-level completion to naturally fill newly revealed disocclusion regions in other views. The stochastic noise introduced during the diffusion forward process generates a perturbed latent $x'_t$. This ensures diversity in the semantics synthesized within these disoccluded regions.

Figure A4 illustrates how the introduced noise leads to semantic diversity in filling disoccluded regions. As the viewpoint moves toward later frames, the downward translation of the chair reveals new regions at the top that must be filled, as indicated by the white regions in the "completion region" of figure A4. Depending on the random seed, different semantics emerge in these newly exposed areas, such as picture frames or hanging plants. This demonstrates the diversity achievable through noise-driven latent-level completion.

Diversity is essential in generative models as it significantly impacts the quality and richness of the generated content. Deterministic approaches often struggle to produce sufficiently varied outputs. This limitation reduces their applicability in scenarios requiring realistic and diverse visual details. By performing completion at the latent level, our method leverages the semantically rich and smooth representation space provided by pretrained diffusion models. Thus, our latent-level approach generates natural and semantically diverse details. This ensures realistic transitions and consistent semantic variation across multiple viewpoints.

## F    Factors affecting visual consistency

In this section, we discuss key factors affecting visual consistency in multi-view image generation. Firstly, employing dense camera trajectories is essential to avoid visual inconsistencies. Attempting to cover wide camera trajectories using a limited number of views results in significant viewpoint differences between frames, causing visual artifacts. Secondly, we observed that setting the classifier-free guidance (CFG) scale too high (approximately 7.5) significantly contributes to flickering artifacts. Thus, selecting an appropriate CFG scale is crucial. An optimal CFG scale of around 5 effectively mitigates flickering while avoiding blurry or noisy images. In conclusion, using denser camera trajectories and carefully choosing the CFG scale significantly reduces visual inconsistencies and improves overall multi-view quality.

## G    Broader impacts

Multi-view content generation and customization are rapidly growing markets, each projected to reach multi-billion-dollar scales by 2030. Currently, commercial AI-driven tools typically focus either on single-view customization or multi-view visualization separately, forcing industries requiring both to depend on expensive and labor-intensive manual workflows. Our proposed multi-view customization framework directly addresses this industrial need by simultaneously enabling balanced performance in customization fidelity and multi-view consistency. By integrating these capabilities, our method can significantly enhance user engagement, sales conversions, and user experiences in fields such as e-commerce, advertising, and virtual reality.

However, as with many generative models, there exists the risk of misuse for malicious or deceptive purposes, such as generating misleading visual content. To mitigate this risk, we restrict our implementation to publicly available, research-focused models that have been released for responsible use. Additionally, our method does not involve training or releasing any models that could produce NSFW or sensitive content, thereby reducing the likelihood of generating harmful material.

