# OpenReview forum: "MVCustom: Multi-View Customized Diffusion via Geometric Latent Rendering and Completion"
_ICLR.cc/2026/Conference — ICLR 2026 Poster_

### Official Review · Reviewer_w9qo · 2025-10-30

**Soundness:** 3
**Presentation:** 3
**Contribution:** 4
**Rating:** 6
**Confidence:** 4

**Summary:**

This paper introduces multi-view customization, a novel task for generating multi-view consistent, customized subjects in new, text-described scenes. The proposed method, MVCustom, uses a video diffusion backbone and two novel inference-stage techniques: DFR for geometric consistency and CLC to realistically fill disoccluded regions.

**Strengths:**

1. Defines a novel, challenging, and high-impact task at the intersection of customization and 3D-aware generation.
2. Presents a highly innovative inference strategy (DFR+CLC) that cleverly enforces geometric consistency on new, text-driven scenes using 2D priors (depth) and feature rendering.
3. Provides compelling experimental validation (especially in the supplementary videos) demonstrating clear superiority over strong baselines in holistic (subject + background) consistency.

**Weaknesses:**

1. Inference Cost / Latency: The proposed inference pipeline is considerably complex, involving multiple forward passes (for CLC) , depth estimation, mesh construction, and differentiable rendering (for DFR). This likely results in significant inference latency compared to standard diffusion models. The paper does not discuss or quantify this computational overhead, which is an important factor for assessing the method's practicality.
2. Rliance on External Depth Estimator: The efficacy of DFR is dependent on the accuracy of an off-the-shelf monocular depth estimator. It is well-known that such estimators can be unreliable on transparent, reflective, or texture-less surfaces. While the paper mentions aligning the depth scale (Appendix B.2), it does not deeply explore the impact of depth estimation failures on the final generation quality.
3. Limitation on Object Pose: As noted in Appendix C, the method cannot alter the subject's intrinsic pose (e.g., from a "sitting" to a "standing" teddy bear) because FeatureNeRF learns a canonical pose from the reference images. This is a reasonable limitation, but given its importance, this discussion should be moved from the appendix to the main paper's limitations section.

**Questions:**

1. What is the quantitative inference latency of the full MVCustom pipeline compared to baselines?
2. How robust is DFR to significant failures from the monocular depth estimator?
3. Can the text prompt influence the subject's intrinsic pose at all, or is it completely fixed by the reference images' canonical pose?

---

> ### Author Response · Authors · 2025-11-22
>
> We sincerely appreciate your detailed review and valuable feedback.
>
> ## Response to weakness1 & question1 : Inference cost / latency
> Thank you for highlighting concerns about inference costs. We acknowledge that MVCustom requires higher computational resources due to the external depth estimator (increasing GPU memory usage) and the feature replacement step (increasing inference time). However, our method comfortably operates on GPUs commonly available today (~25 GB memory, e.g., NVIDIA A6000), and the increased inference time remains practically manageable. Furthermore, explicitly enforcing geometric consistency at inference is critical given the constraint of extremely limited training data.
>
> We report detailed per-sample latency and peak memory usage for multi-view customization below:
>
> | Method                  | Inference Time (s) | GPU Memory (GB) | Camera Pose Accuracy (↑) | Multi-view Consistency (↓) | Identity Preservation (↓) | Text Alignment (↑) |
> | ----------------------- | ------------------ | --------------- | ------------------------ | -------------------------- | ------------------------- | ------------------ |
> | Custom Img + Img-MV gen | *96.18*            | *6.73*          | *0.675 ± 0.123*          | 0.214 ± 0.145              | 0.504 ± 0.124             | 0.676 ± 0.105      |
> | Txt-MV gen with DB      | **27.20**          | **5.42**        | 0.283 ± 0.254            | **0.116 ± 0.085**          | 0.557 ± 0.121             | 0.723 ± 0.095      |
> | CustomDiffusion360      | 74.97              | 4.99            | 0.000 ± 0.000            | 0.190 ± 0.107              | **0.417 ± 0.115**         | **0.806 ± 0.102**  |
> | MVCustom (Ours)         | 130.92             | 19.29           | **0.735 ± 0.101**        | *0.121 ± 0.104*            | *0.448 ± 0.112*           | *0.744 ± 0.104*    |
>
> (*Best results in bold, second-best in italics.*)
>
>
> We argue that MVCustom's significant improvements in multi-view consistency, geometric accuracy, and customization fidelity clearly justify the computational trade-off.
> We revised our manuscript to explicitly include this inference cost analysis  in section 4.2 Results - Multi-view consistency with perspective alignment. (revisions highlighted in blue).

---

> ### Author Response · Authors · 2025-11-22
>
> ## Response to weakness 2 & question 2: Reliance on external depth estimator
> We acknowledge that inaccuracies in depth estimation can negatively impact the geometric realism of backgrounds across views, even if consistency is maintained. For example, reflective surfaces such as mirrors may incorrectly be perceived as distant, causing unrealistic separations under viewpoint changes. We respectfully note that this limitation originates primarily from the external depth estimator rather than from our framework itself, and similar issues are common in other depth-conditioned methods[1,2,3].
>
> Recent advancements in depth estimation models[4,5] have substantially improved accuracy on reflective and textureless surfaces, offering significant potential to mitigate these issues. To further address reviewers' concerns, we provide additional evidence in Figure 6 of the revised manuscript, clearly demonstrating that correcting depth estimation inaccuracies substantially enhances background realism. Thus, we anticipate that ongoing improvements in depth estimation techniques will soon alleviate this limitation, further enhancing our framework's robustness.
>
> We explicitly discuss this limitation and potential improvements in our revised manuscript (Section 5, highlighted in blue).
>
> Thanks to your valuable feedback, we recognize that adopting more robust depth estimators for challenging cases, such as textureless surfaces, can significantly reduce failure scenarios. Accordingly, in the camera-ready version, we will update our results using the improved depth estimator, DepthAnything2[4].
>
>
> ## Response to weakness 3 & question 3: Limitation on reference object pose
> We agree that our method currently cannot alter the intrinsic object pose purely through text conditions. This limitation arises because FeatureNeRF learns a fixed canonical pose from the reference images without conditioning on textual input. As suggested, we included this limitation in the revised main manuscript (Section 5, revisions highlighted in blue).
>
>
> Additionally, we clarify the degree to which object shape (though not intrinsic pose) can be controlled by adjusting the usage ratio of pose-conditioned transformer blocks (containing FeatureNeRF) during denoising steps. We provide detailed examples demonstrating these variations in *new supplementary HTML file: rebuttal_mvcustom.html, Section C. Shape variants with different pose-conditioned transformer block usage ratios in inference.*):
>
> * **Case (a): Using pose-conditioned transformer blocks in 100% of the denoising steps**
> Accurately maintains viewpoint alignment, preserves the reference object's shape, and partially reflects shape variations specified by text.
>
> * **Case (b): Using pose-conditioned transformer blocks in 75% of the denoising steps**
>   Allows greater reflection of text-specified shape variations, slightly compromising reference shape consistency.
>
> * **Case (c): Using pose-conditioned transformer blocks in 50% of the denoising steps**
>   Reflects detailed shape variations from text prompts while partially preserving key reference features (e.g., color or door shape), but viewpoint alignment begins to degrade.
>
> * **Case (d): Using pose-conditioned transformer blocks in 25% of the denoising steps**
>   Does not retain reference object details or viewpoint consistency, only accurately reflecting the general object type and details described by the text. Hence, this configuration does not constitute meaningful reference customization.
>
> Intrinsic pose variations (e.g., changing from sitting to standing) cannot be achieved simply by adjusting pose-conditioned transformer usage. Such poses are fixed primarily in early denoising steps; omitting pose conditioning in these early steps completely disrupts viewpoint alignment, despite potentially enabling text-driven pose variations.
>
> We greatly appreciate your insightful feedback. We will provide user-adjustable timestep control in our released code.
>
>
> ### Reference
> [1] Unified Dense Prediction of Video Diffusion
> [2] Training-free Camera Control for Video Generation
> [3] IDCNet: Guided Video Diffusion for Metric-Consistent RGBD Scene Generation with Precise Camera Control
> [4] Depth Anything V2
> [5] DepthFocus: Controllable Depth Estimation for See-Through Scenes

---

### Official Review · Reviewer_yETe · 2025-10-30

**Soundness:** 3
**Presentation:** 3
**Contribution:** 3
**Rating:** 6
**Confidence:** 2

**Summary:**

This paper presents MVCUSTOM, a  diffusion-based framework explicitly designed
to achieve both multi-view consistency and customization fidelity. Specifically, they propose a novel task, multi-view customization, and  address multi-view consistency and limited data problem in customization.

**Strengths:**

• The paper is well-written with a logical structure that makes the technical contributions easy to follow.
• The proposed framework integrating dense spatio-temporal attention modules into video diffusion-based backbone is well-justified and addresses limitations in multi-view consistency via temporal coherence.

**Weaknesses:**

1The author only conduct experiments on a U-Net-based video diffusion model, I wonder if the proposed method can further improve performance on DiT-based models.

Suggestions:
1.	Conduct experiments on DiT-based backbone like Wan2.1?
2.	Include a comprehensive supplementary video showing: Side-by-side comparisons with all baselines (at least 3-5 examples per method)
3.	Add more dense camera pose trajectory to show more consisent.
4.	Create a project page with interactive demos which allow users to adjust the camera parameters.

**Questions:**

see weakness

---

> ### Author Response · Authors · 2025-11-22
>
> We sincerely appreciate your detailed review and valuable suggestions.
>
> ## Response to suggestion 1: Experiments on DiT-based video backbone
> We agree that adopting a DiT-based video diffusion backbone may potentially enhance visual quality. However, our decision to use a U-Net based backbone, *despite accepting some loss in video quality*, was driven by the *need for precise geometry learning and effective camera pose control under extremely limited data conditions*.
>
> Recent DiT-based models typically require additional conditioning networks to achieve frame-level camera pose control. Such networks demand large-scale datasets[1,2,3]: for example, RelCamMaster uses 136K videos, and ACD uses 65M videos. In contrast, our multi-view customization setting provides only a single short video (~50 frames), insufficient for reliably training these networks without severe overfitting or losing viewpoint controllability.
>
> Furthermore, DiT architectures compress temporal dimensions via a 3D VAE, making it difficult to maintain explicit per-frame feature maps needed for precise camera pose alignment. The U-Net based AnimateDiff model, however, uses a 2D VAE, explicitly preserving per-frame latent feature maps. Leveraging this property, we directly enforce explicit 3D geometry constraints through FeatureNeRF and Feature Mesh Rendering, effectively ensuring accurate viewpoint control and geometric consistency even with extremely limited training data.
>
>
> ## Response to suggestions 2 and 3: Video comparison and dense camera trajectory
> We acknowledge that the initial supplementary material, which used only 8 frames for wide camera trajectories, was insufficient. Following your valuable suggestions, we now provide additional comparisons using denser camera trajectories (increased frame counts along the same trajectory), resulting in significantly improved visual consistency.
> Please refer to *Section A of our new supplementary HTML file: rebuttal_mvcustom.html* for detailed video results.
>
> ## Response to suggestion 4: Interactive demo
> We fully agree that providing an interactive demo is important for user convenience. We will implement an interactive demo for the inference stage, where users can interactively select target camera trajectories using a simple UI, conditioned on pre-uploaded customized model checkpoints. Due to practical constraints (fine-tuning a customized model typically takes around one day), the demo will utilize pre-trained checkpoints.
>
> We will prepare and release this interactive demo by the camera-ready deadline.
>
> ### Reference
> [1] ReCamMaster: Camera-Controlled Generative Rendering from A Single Video
> [2] AC3D: Analyzing and Improving 3D Camera Control in Video Diffusion Transformers
> [3] SynCamMaster: Synchronizing Multi-Camera Video Generation from Diverse Viewpoints

---

### Official Review · Reviewer_QASA · 2025-10-31

**Soundness:** 3
**Presentation:** 3
**Contribution:** 3
**Rating:** 6
**Confidence:** 2

**Summary:**

This paper introduces a diffusion-based framework named MVCustom, which is designed for multi-view customization with explicit camera pose control. For the training stage, it employs a video-diffusion backbone with spatio-temporal attention. To address the challenge of limited training data during inference, the paper proposes two novel techniques: depth-aware feature rendering to enforce geometric consistency and consistent-aware latent completion to realistically synthesize content in newly revealed regions.

**Strengths:**

1. The paper defines a novel task of multi-view customization that combines camera control with subject personalization.
2. The proposed inference-time techniques for ensuring multi-view consistency effectively address the challenge of limited training data.
3. The paper is well-structured and easy to follow.

**Weaknesses:**

1. Could you provide more details on the inference time of MVCustom compared to the baselines? The inference process, particularly the depth-aware feature rendering component, appears to be computationally intensive.
2. On the project page, some results still show obvious hallucinations or inconsistencies when the viewpoint variation is large. Is there further analysis on this specific limitation? A discussion or visualization of typical failure cases would be helpful to understand the method's boundary conditions better.

**Questions:**

Please refer to the weaknesses.

---

> ### Author Response · Authors · 2025-11-22
>
> We sincerely appreciate your thorough review and valuable feedback.
>
> ## Response to weakness 1 : regarding inference costs:
> We acknowledge that MVCustom requires higher computational resources due to the external depth estimator (increasing GPU memory usage) and the feature replacement step (increasing inference time). However, our method comfortably operates on GPUs commonly available today (~25 GB memory, e.g., NVIDIA A6000), and the increased inference time remains practically manageable. Furthermore, explicitly enforcing geometric consistency at inference is critical given the constraint of extremely limited training data.
>
> We report detailed per-sample latency and peak memory usage for multi-view customization below:
>
> | Method                  | Inference Time (s) | GPU Memory (GB) | Camera Pose Accuracy (↑) | Multi-view Consistency (↓) | Identity Preservation (↓) | Text Alignment (↑) |
> | ----------------------- | ------------------ | --------------- | ------------------------ | -------------------------- | ------------------------- | ------------------ |
> | Custom Img + Img-MV gen | *96.18*            | *6.73*          | *0.675 ± 0.123*          | 0.214 ± 0.145              | 0.504 ± 0.124             | 0.676 ± 0.105      |
> | Txt-MV gen with DB      | **27.20**          | **5.42**        | 0.283 ± 0.254            | **0.116 ± 0.085**          | 0.557 ± 0.121             | 0.723 ± 0.095      |
> | CustomDiffusion360      | 74.97              | 4.99            | 0.000 ± 0.000            | 0.190 ± 0.107              | **0.417 ± 0.115**         | **0.806 ± 0.102**  |
> | MVCustom (Ours)         | 130.92             | 19.29           | **0.735 ± 0.101**        | *0.121 ± 0.104*            | *0.448 ± 0.112*           | *0.744 ± 0.104*    |
>
> (*Best results in bold, second-best in italics.*)
>
> We argue that MVCustom's significant improvements in multi-view consistency, geometric accuracy, and customization fidelity clearly justify the computational trade-off.
> We revised our manuscript to explicitly include this inference cost analysis at section 4.2 Results - Multi-view consistency with perspective alignment. (revisions highlighted in blue).
> ### Response to weakenss 2: some results on inconsistency when large viewpoint variation
> Thank you for thoroughly examining our video results and pointing out this critical issue.
> We identified two main factors and corresponding solutions for the observed inconsistencies:
>
> 1. Dense camera trajectory:
> The sparse frame count (8 frames) over wide trajectories caused minimal overlap between frames, leading to visual artifacts. Based on your suggestion, we generated denser trajectories with increased frame counts (16 frames), resulting in significantly improved consistency. Please see *Section A of our new supplementary HTML file: rebuttal_mvcustom.html*.
>
> 2. CFG scale adjustment:
> We found that a larger CFG scale (previously set around 7.5) contributed to flickering. Adjusting the CFG scale to a lower, optimal value substantially reduced flickering and improved visual consistency. See examples in *Section B of our new supplementary HTML file: rebuttal_mvcustom.html.*
>
> In summary, dense camera trajectories and appropriate CFG scale settings significantly improve visual consistency.
> Since these factors notably influence multi-view generation quality, we have added a dedicated discussion in Appendix F of our revised manuscript to support future research.

---

### Official Review · Reviewer_azR9 · 2025-10-31

**Soundness:** 2
**Presentation:** 2
**Contribution:** 2
**Rating:** 4
**Confidence:** 2

**Summary:**

This manuscript presents a joint multi-view generation with explicit camera pose control and subject customization task, and a corresponding framework to solve this task. More specifically, the authors proposed an MVCustom framework, which based on a diffusion model to generate multi-view images controlled by explicit camera poses with a consistent subject from given reference images. MVCustom is based on AnimateDiff and incorporates an additional feature NeRF and dense spatio-temporal layers to achieve multi-view consistency and subject consistency. In addition, the authors also introduce a depth-aware feature rendering to better achieve geometric consistency, and a consistent-aware latent completion technique to ensure the appearance consistency of both the subject and the surroundings.

**Strengths:**

The newly proposed task of jointly performing subject customization and multi-view image generation is a direction that has been rarely explored.

**Weaknesses:**

1. The reviewer is concerned about the practical application value and novelty of the proposed multi-view customization task. This task can be framed as a sparse-view version of controllable (camera pose-conditioned) video customization. The paper’s own choice of a video diffusion backbone (AnimateDiff) and its method of "enhancing temporal coherence into holistic-frames consistency" further reinforces this similarity. The practical distinction and added value over existing controllable video generation or customization frameworks are not sufficiently justified.

2. The framework is built on a U-Net-based video diffusion model (AnimateDiff), which is arguably dated compared to recent transformer-based architectures. While the authors justify this in the appendix (Sec. A.1) as a compatibility choice with FeatureNeRF and frame-level precise camera control, the quality of the generated images is still highly constrained by the base model. Meanwhile, the core mechanism for camera control (pose-conditioned transformer block) is admittedly adopted from a previous work, further limiting its novelty. The other two main contributions, depth-aware feature rendering and consistent-aware latent completion, are presented as inference-stage techniques but lack rigorous quantitative validation.

3.	As for the quantitative results shown in Tab. 2, the quantitative results in Table 2 do not show clear and comprehensive superiority, but a mixed trade-off rather than a decisive win, making the paper's claim of being the "only one that achieves consistently strong performance" an overstatement. Meanwhile, while the figures (e.g., Fig. 4) show improvements over baselines, inconsistencies are still apparent. This is particularly noticeable in the supplementary videos, where flickering and geometric instability suggest that holistic multi-view consistency is not fully solved, especially for the synthesized backgrounds that are not geometrically constrained by the FeatureNeRF model (explicit 3d modeling). To more robustly evaluate the claimed multi-view and geometric consistency, the authors should attempt to reconstruct an explicit 3D model (e.g., a mesh or Neural Radiance Field) from the generated multi-view outputs. This would provide a much stronger and more objective measure of 3D consistency than 2D metrics.

4. The ablation study in Section 4.3 is insufficient as it is purely qualitative. The paper introduces several key techniques (Depth-aware Feature Rendering, Consistent-aware Latent Completion) but fails to provide any quantitative analysis to demonstrate their individual impact on the final metrics (e.g., in Table 2). Figure 5 offers only a single visual comparison, which is not enough to fully validate the effectiveness of each proposed module.

**Questions:**

Please refer to the weaknesses.

---

> ### Author Response · Authors · 2025-11-24
>
> We sincerely thank Reviewer azR9 for the detailed and thoughtful review.
>
> ## Response to weakness 1: Practical application value and novelty of multi-view customization
> Both multi-view content generation and content customization are rapidly expanding markets, each projected to reach multi-billion dollar scales by 2030 [1,2]. Integrating 3D visuals significantly increases user engagement and sales conversions, as demonstrated by Shopify’s 94% sales increase and marketing campaigns showing conversion improvements over 40% [3,4].
> However, current AI-driven commercial tools typically support either single-view customization (e.g., Google's Nano Banana) or multi-view visualization (e.g., Cappasity) [5], but not both simultaneously. Consequently, industries requiring both capabilities rely on costly and labor-intensive manual processes using tools such as Unreal Engine [6,7].
>
> Thus, multi-view customization directly bridges these two distinct markets, addressing a clear and growing industrial need. Importantly, multi-view customization is not merely an extension of existing tasks. Rather, it introduces a novel challenge requiring balanced performance across both customization fidelity and multi-view consistency. Due to this inherent difficulty, existing AI-driven tools have so far been specialized separately for each task, without successfully extending to a combined multi-view customization scenario. Simply extending existing methods, as demonstrated by the competitor "Txt-MV gen with DB" (applying a conventional customization method onto CameraCtrl), results in poor customization performance and significant degradation in camera controllability (Table 2, Table A3, qualitative examples).
>
> Thank you for raising concerns regarding the practical value and novelty of multi-view customization. We revised our manuscript to clarify the broader implications of this discussion in Appendix E (Broader Impact), with revisions highlighted in blue.
>
> ## Response to weakness 2 : Justification for using a U-Net backbone instead of DiT-based backbone
>
>
> We acknowledge that adopting a U-Net based video diffusion backbone may limit visual quality compared to recent transformer based architectures. Nevertheless, we intentionally chose a U-Net architecture, *accepting some loss in video quality*, because *our primary goal is accurate geometry learning and effective frame-level control under extremely limited data conditions*.
>
> Recent DiT based video models typically require training an additional conditioning network for precise frame-level camera pose control[8,9,10]. However, these approaches rely heavily on large-scale datasets ; RelCamMaster uses 136K videos and ACD 65M videos. By contrast, our multi-view customization scenario provides only a single video sample (about 50 frames), insufficient to stably train such additional networks without severe overfitting and loss of controllability.
>
> Therefore, recognizing the fundamental limitations of DiT based architectures under extremely limited data, we adopted a U-Net architecture that maintains per frame latent features. This enables us to explicitly impose geometry constraints at inference using Feature Mesh Rendering, thus achieving robust viewpoint controllability and geometric consistency.

---

> ### Author Response · Authors · 2025-11-24
>
> ### Reference 1.
>
>
> [1] Photo Editing Software Market Growth and Trends, Research and Markets. (Market Snapshot)
> [https://www.researchandmarkets.com/report/image-editing-software](https://www.researchandmarkets.com/report/image-editing-software)
>
> [2] 3D Digital Asset Market Summary, Grand View Research.
> [https://www.grandviewresearch.com/industry-analysis/3d-digital-asset-market-report](https://www.grandviewresearch.com/industry-analysis/3d-digital-asset-market-report)
>
> [3] Apviz report
> [https://apviz.io/blog/3d-marketing](https://apviz.io/blog/3d-marketing)
>
> [4] "Higher conversion rates" section: Impact of 3D visuals on e-commerce sales.
> [https://www.shopify.com/blog/3d-ecommerce](https://www.shopify.com/blog/3d-ecommerce)
>
> [5] Importance of 3D marketing in the manufacturing industry, highlighting 82% adoption of 3D viewers by online shoppers.
> [https://apviz.io/blog/3d-marketing/#the-importance-of-3d-marketing-strategy-in-the-manufacturing-industry](https://apviz.io/blog/3d-marketing/#the-importance-of-3d-marketing-strategy-in-the-manufacturing-industry)
>
> [6] TODAY Home: IKEA uses CGI to create multi-view furniture catalog images.
> [https://www.today.com/home/ikea-reveals-75-catalog-images-are-cgi-1d80127051](https://www.today.com/home/ikea-reveals-75-catalog-images-are-cgi-1d80127051)
>
> [7] Audi and Mackevision create digital showrooms using Unreal Engine.
> [https://www.unrealengine.com/en-US/spotlights/creating-a-digital-showroom-audi-and-mackevision-choose-ue4](https://www.unrealengine.com/en-US/spotlights/creating-a-digital-showroom-audi-and-mackevision-choose-ue4)
>
> [8] ReCamMaster: Camera-Controlled Generative Rendering from A Single Video
> [9] AC3D: Analyzing and Improving 3D Camera Control in Video Diffusion Transformers
> [10] SynCamMaster: Synchronizing Multi-Camera Video Generation from Diverse Viewpoints

---

> ### Author Response · Authors · 2025-11-24
>
> ## Response to weakness 3 : Evaluation
>
> #### 1. Response to concern about "No decisive win": Balanced multi-view consistency and customization fidelity
>
> We emphasize that balanced high performance across multi-view consistency and customization fidelity is more valuable than narrowly excelling in a single metric. Unlike our specialized baseline competitors, MVCustom consistently performs well across all metrics, achieving the lowest mean rank and smallest rank standard deviation, as shown in the table below:
>
> | Method                  | Mean Rank | Std of Rank | Camera Pose Accuracy Rank | Multi-view Consistency Rank | Identity Preservation Rank | Text Alignment Rank |
> | ----------------------- | --------- | ----------- | ------------------------- | --------------------------- | -------------------------- | ------------------- |
> | Custom Img + Img-MV gen | 3.25      | 0.96        | 2                         | 4                           | 3                          | 4                   |
> | Txt-MV gen with DB      | 2.75      | 1.26        | 3                         | 1                           | 4                          | 3                   |
> | CustomDiffusion360      | 2.25      | 1.50        | 4                         | 3                           | 1                          | 1                   |
> | MVCustom (Ours)         | **1.75**  | **0.50**    | 1                         | 2                           | 2                          | 2                   |
>
> We have clarified this in our revised manuscript:
>
> > **Before:** "MVCustom is the only framework that simultaneously achieves faithful multi-view generation and customization."
>
> > **After:** "MVCustom achieves balanced and consistently competitive performance across multi-view consistency, customization fidelity, and visual quality, effectively addressing the multi-objective generation task."
>
>
> Practically, consistently achieving top-two performance across multiple metrics is more valuable than excelling narrowly in a single metric. Thus, our balanced results reflect effective management of multiple competing objectives rather than a "mixed trade-off."
>
>
>
>
> #### 2. Mitigating Flickering via Dense Trajectories and CFG Scale
>
> We observed that sparse viewpoints across wide trajectories result in inconsistencies and visual artifacts. To address this, we increased the frame count from 8 to 16, creating denser trajectories that significantly reduce artifacts (please refer to *Section A of our new supplementary HTML file: rebuttal_mvcustom.html*). Additionally, lowering the CFG scale from 8 to 5 further mitigates flickering (please refer to *Section B of our new supplementary HTML file: rebuttal_mvcustom.html*). Thus, employing dense camera trajectories and optimal CFG scale substantially improves visual consistency.
>
> #### 3. Additional Evaluation of 3D Consistency: COLMAP Reconstruction
>
> We adopted COLMAP reconstruction to quantitatively assess geometric consistency, following established protocols [1,2].
>
> | Method                  | # of point cloud by COLMAP (↑) | Camera Pose Accuracy (↑) | Multi-view Visual Similarity (↓) | Identity Preservation (↓) | Text Alignment (↑) |
> | ----------------------- | ------------------------------ | ------------------------ | -------------------------------- | ------------------------- | ------------------ |
> | Custom Img + Img-MV gen | **45.286**                     | *0.675*                  | 0.216                            | 0.504                     | 0.676              |
> | Txt-MV gen with DB      | 12.586                         | 0.283                    | **0.116**                        | 0.557                     | 0.723              |
> | CustomDiffusion360      | 0                              | 0.000                    | 0.190                            | **0.417**                 | **0.806**          |
> | MVCustom (Ours)         | *44.951*                       | **0.735**                | *0.121*                          | *0.448*                   | *0.744*            |
>
> (Best results are highlighted in bold; second-best in italic. COLMAP points indicate the average number of reconstructed points from multi-view images with target camera poses, generated from two concepts using evaluation prompts and 16 frames.)
>
> MVCustom achieves competitive 3D consistency and maintains balanced performance across metrics. Custom Img + Img-MV gen achieves high COLMAP points but loses customization fidelity at larger viewpoints (as shown by supplementary video results, low identity preservation, and text alignment scores). CustomDiffusion360 and Txt-MV gen fail reconstruction due to background randomness and pose inaccuracies, respectively.
>
> Additionally, metrics like PSNR or LPIPS using optimized NeRF or 3DGS are unsuitable for evaluating multi-view customization, as our outputs rely solely on textual conditions rather than ground truth images.

---

> > ### Author Response · Authors · 2025-11-24
> >
> > ### Reference 2.
> > [1] DiffDreamer: Towards Consistent Unsupervised Single-view Scene Extrapolation with Conditional Diffusion Models
> > [2] EG3D: Efficient Geometry-aware 3D Generative Adversarial Networks

---

> ### Author Response · Authors · 2025-11-24
>
> ## Response to weakness 4: Quantitative validation and additional qualitative samples on inference stage
>
> We provide additional qualitative ablation samples in *Fig. A3 of the new supplementary Appendix*:
>
> * **Only model customization (a)** aligns the target camera pose only with the customized object.
> * **Adding depth-aware feature rendering (b)** aligns the background to the target camera pose but retains previous content in disoccluded regions.
> * **Adding consistent-aware latent completion (c)** generates new content in disoccluded regions, significantly enhancing realism.
>
> Moreover, latent completion ensures diversity in newly generated disocclusion regions, as demonstrated in Appendix E.
>
> We also provide quantitative ablation results, highlighting the significant contribution of our inference strategy, particularly depth-aware feature rendering, to geometric-aware consistency:
>
> | Method                                           | # COLMAP recon (↑) | PoseAcc (↑)       | Multi-view consistency (↓) | Identity Preservation (↓) | Text alignment (↑) |
> | ------------------------------------------------ | ------------------ | ----------------- | -------------------------- | ------------------------- | ------------------ |
> | **(a)** Only model customization                 | 36.13 ± 19.87      | 0.543 ± 0.179     | *0.095 ± 0.067*            | 0.355 ± 0.076             | **0.682 ± 0.074**  |
> | **(b)** (a) + Depth-aware feature rendering      | *43.38 ± 15.98*    | *0.768 ± 0.153*   | **0.090 ± 0.065**          | **0.347 ± 0.068**         | 0.679 ± 0.073      |
> | **(c)** (b) + Consistent-aware latent completion | **45.38 ± 26.39**  | **0.771 ± 0.142** | 0.113 ± 0.081              | *0.384 ± 0.086*           | *0.681 ± 0.066*    |
>
> (Best results are highlighted in bold; second-best in italic. Evaluations conducted on rotation-aware camera trajectory and x-y translation trajectory.)
>
> The slight reduction in multi-view consistency scores after adding consistent-aware latent completion reflects the introduction of diverse new content rather than visual inconsistency, as confirmed by COLMAP reconstruction results and qualitative examples.
>
> We appreciate your valuable feedback, which has significantly enhanced our manuscript. This discussion is revised in Appendix H (revisions highlighted in blue) .

---

### Author Response · Authors · 2025-11-28

As we approach the final stage of the discussion period, **we warmly invite reviewers to engage further with our discussion.** For your convenience, please refer to the updated supplementary material (rebuttal_mvcustom.html).

Summary of overlapping feedback from reviewers :
* Dense trajectories and CFG scale adjustment significantly improve visual consistency (Sections A and B of the supplementary).
[reviewer QASA, weakness 2 / reviewer azR9, weakness 3 / reviewer yETe, suggestion 3]

* Inference cost: Our inference takes only 30 seconds and two minutes longer than the slowest and fastest baselines, respectively, which are still okay for the users. GPU memory usage (19.29 GB) is compatible with commonly available ~24 GB GPUs such as RTX 3090.
[reviewer QASA, weakness 1 / reviewer w9qo, weakness 1]


* U-Net backbone choice: Despite slightly lower visual quality compared to DiT-based backbones, the U-Net architecture enables explicit 3D modeling in the latent space because it preserves the temporal axis while DiT compresses eight frames to one, crucial for reliable geometry control with limited data.
[reviewer azR9, weakness 2 / reviewer yETe, suggestion 1]

We have carefully incorporated reviewers’ feedback into the revised manuscript. If any further clarification is needed, please feel free to leave another comment.
We sincerely appreciate the reviewers for recognizing the strengths of our work, specifically:
* The paper proposes a novel and impactful task combining customization, multi-view consistency, and camera control, which is seen as timely, useful, and underexplored.
* The inference-time strategy is regarded as innovative and effective for enforcing geometric and holistic consistency under limited data.
* Experiments show clear superiority over strong baselines, supported by thorough and well-justified analyses and ablations.

---

### Author Response · Authors · 2025-12-03
**Summary for Area Chairs 1/2**

We sincerely appreciate the Area Chairs’ efforts in maintaining the integrity of the review process.
The reviewers initially rated our paper as
>4 marginally below the acceptance threshold [azR9]

> 6 marginally above the acceptance threshold [QASA]

> 6 marginally above the acceptance threshold [yETe]

> 6 marginally above the acceptance threshold [w9qo]

and asks potential improvements as follows.

Below, we summarize our paper and address key issues raised by reviewers.

## Paper summary
- This paper introduces **multi-view customization**, a novel task of generating customized subjects in new, text-described scenes while ensuring explicit camera pose control and holistic multi-view consistency for both the subject and its surroundings.
- The proposed method, **MVCustom**, uses a video diffusion backbone and two inference-stage techniques: **Depth-aware Feature Rendering (DFR)** for geometric consistency and **Consistent-aware Latent Completion (CLC)** to realistically fill disoccluded regions.
- MVCustom significantly outperforms brute-force approaches such as sequential customization-then-multi-view methods ([DreamBooth LoRA](https://github.com/huggingface/diffusers/blob/main/examples/advanced_diffusion_training/train_dreambooth_lora_sd15_advanced.py) + CameraCtrl [He et al., 2024], single customized image conditioned on SEVA [Zhou et al., 2025]), and CustomDiffusion360[Kumari et al., 2024], which excludes a consistent background from its task.

---

> ### Author Response · Authors · 2025-12-03
> **Summary for Area Chairs 2/2**
>
> ## Common issues raised by reviewers and our resolutions
> **Novelty of the proposed task**
> - Positive assessment
>   - Reviewer *w9qo* describes our work as “a novel, challenging, and high-impact task at the intersection of customization and 3D-aware generation,” and reviewer *QASA* similarly identifies it as “a novel task of multi-view customization. Also, reviewer *azR9* agrees that the proposed problem "has been rarely explored."
> - Concern
>   - Reviewer *azR9*  raises the concern that the task might be addressed by applying customization to existing camera pose-controllable video generation models.
>     - **Our response:** Directly integrating existing methods (e.g., our competitor "Txt-MV gen with DB") significantly underperforms in multi-view customization. In the experiments, this competitor achieves a reference-subject fidelity of 0.557 (lower is better), 0.109 worse (19.6% relative decrease) than MVCustom's 0.448, the lowest fidelity among baselines (Table 2). It also reduces camera pose controllability, increasing rotation error by 5.7% (from 15.660 to 16.550) and translation error by 5.1% (from 4.385 to 4.608), as detailed in Table A3. These results justify the need for a dedicated framework.
> ---
> **Inference-Time strategy and inference latency**
> - Positive assessment
>   - Reviewer *QASA*: “The proposed inference-time techniques effectively address *the challenge of limited training data.*”
>   - Reviewer *w9qo*: "Presents a highly innovative inference strategy (DFR+CLC)"
> - Concern
>   - Reviewers *QASA*, *azR9*, and *w9qo* request explicit comparisons of inference time and GPU memory usage.
>     - **Our response:** MVCustom remains feasible on commonly available ~24 GB GPUs such as the RTX 3090. Specifically, it requires approximately 30 seconds to two minutes more inference time than the slowest and fastest baselines and uses 19.29 GB of GPU memory.
>   - Reviewer *azR9* also notes that quantitative and additional qualitative ablations are necessary.
>     - **Our response:** DFR significantly improves holistic scene geometry as supported by the quantitative ablations in the revised manuscript (Appendix H): the number of COLMAP reconstruction points increases by 20.1% (36.13→43.38), and camera pose accuracy improves by 41.4% (0.543→0.768). Additional qualitative examples further confirm that CLC adds realistic content in newly visible regions.
> ---
> **Comparisons and consistency under large viewpoint changes**
> - Positive assessment
>   - Reviewer *w9qo* emphasizes that the “video comparison demonstrates clear superiority over strong baselines in holistic (subject + background) consistency.”
> - Concerns
>   - Reviewers *azR9* points out that some results have flickering and inconsistencies. And reviewer *yETe* suggests using denser camera pose trajectories to improve consistency.
>     - **Our response:** Denser camera trajectories (more frames along the same trajectory) significantly improve consistency, as shown in Section A of the new supplementary (`rebuttal_mvcustom.html`). It implies that the previous artifacts were largely due to the sparse trajectories rather than the framework.
>     - Competitors still show inconsistent surroundings or failed customization under the same conditions.
>   - Reviewers *QASA* request further analysis of the flickering and hallucination case.
>     - **Our response:** An analysis on CFG scales helped mitigate the flickering and hallucination as shown in Section B of the `rebuttal_mvcustom.html`. The CFG scale is now set to 5.
>   - Reviewer *azR9* notes that although MVCustom ranks first or second across multi-view and customization metrics, it does not achieve the best score in every metric.
>       - **Our response:** Balancing high performance across multiple metrics is practically essential for multi-objective tasks because failure in either customization or multi-view generation leads to unusable results. Our rank-based analysis demonstrates that competing methods excel only in isolated metrics, resulting in inferior mean ranks (2.25–3.25) and variances (0.96–1.50). In contrast, MVCustom achieves the best mean rank (1.75) and smallest variance (0.50), confirming balanced and robust performance.
> ---
>
> The revised manuscript and updated supplementary (rebuttal_mvcustom.html) comprehensively address reviewer feedback, clearly discussing practical applications, ablation analyses, and limitations.
>
> We appreciate the reviewers’ valuable feedback and had hoped to further clarify and highlight these points during the reviewer–author discussion. Unfortunately, the discussion phase ended before meaningful interaction could occur, despite our timely submission of all revisions. While understandable, this means *our submission did not benefit from the reviewer–author interaction phase*. We respectfully request that the AC consider the rebuttal, supplementary analyses, and revised manuscript in forming the final recommendation.

---

### Meta-Review · Area_Chair_AAiu · 2026-01-07

**Summary:**

This paper introduces multi-view customization, an interesting task that unifies camera pose control with subject customization under multi-view consistency requirements.

Reviewers generally found the task well-motivated and the proposed MVCustom framework technically sound. Initial concerns regarding task novelty, inference efficiency, limited quantitative ablations, and visual inconsistencies under large viewpoint changes were substantially addressed in the rebuttal.

The authors clarified the difference from prior controllable video and customization settings, added quantitative ablations and geometric consistency evaluations, reported inference cost and memory usage, and analyzed failure cases.

Overall, the revised submission demonstrates solid technical contributions and credible experimental validation. I recommend acceptance as a poster.

**Reviewer Concerns:**

Concerns addressed:

1. Task novelty clarified with empirical evidence showing naïve combinations of existing methods fail.
2. Quantitative ablations added for inference-stage components, including COLMAP-based geometric evaluation.
3. Inference time and GPU memory usage reported and shown to be feasible.
4. Flickering and instability analyzed and mitigated via denser trajectories and CFG tuning.

Remaining:
1. Use of a U-Net backbone may limit visual quality compared to DiT-based models, but is reasonably justified under limited-data constraints.
2. Performance is balanced across metrics rather than best-in-class on all individual metrics.

**Reviewer Scores:**

Reviewer azR9:
Likely to keep reject 4. He/she concerned about the practical application value and novelty.

Reviewer QASA:
Likely to remain at 6, with increased confidence. The rebuttal fully addressed inference cost questions and provided concrete explanations and analyses for observed failure cases.

Reviewer yETe:
Likely to remain at 6. The discussion on denser trajectories and CFG tuning directly responds to suggestions, though no explicit score change was indicated.

Reviewer w9qo:
Likely unchanged at 6, as this reviewer was already positive and explicitly praised the inference-time strategy and holistic consistency.

---

### Decision · Program_Chairs · 2026-01-26

Accept (Poster)